# Co-targeting myelin inhibitors and CSPGs markedly enhances regeneration of GDNF-stimulated, but not conditioning-lesioned, sensory axons into the spinal cord

Jinbin Zhai[1,2], Hyukmin Kim[1,2], Seung Baek Han[1,2], Meredith Manire[1,2], Rachel Yoo[1,2], Shuhuan Pang[1,2], George M Smith[1,2], Young-Jin Son[1,2]*

[1]Shriners Hospitals Pediatric Research Center and Center for Neural Repair and Rehabilitation, Lewis Katz School of Medicine, Temple University, Philadelphia, United States; [2]Center for Neural Repair and Rehabilitation, Lewis Katz School of Medicine, Temple University, Philadelphia, United States

*For correspondence:
yson@temple.edu

**Abstract** A major barrier to intraspinal regeneration after dorsal root (DR) injury is the DR entry zone (DREZ), the CNS/PNS interface. DR axons stop regenerating at the DREZ, even if regenerative capacity is increased by a nerve conditioning lesion. This potent blockade has long been attributed to myelin-associated inhibitors and (CSPGs), but incomplete lesions and conflicting reports have prevented conclusive agreement. Here, we evaluated DR regeneration in mice using novel strategies to facilitate complete lesions and analyses, selective tracing of proprioceptive and mechanoreceptive axons, and the first simultaneous targeting of Nogo/Reticulon-4, MAG, OMgp, CSPGs, and GDNF. Co-eliminating myelin inhibitors and CSPGs elicited regeneration of only a few conditioning-lesioned DR axons across the DREZ. Their absence, however, markedly and synergistically enhanced regeneration of GDNF-stimulated axons, highlighting the importance of sufficiently elevating intrinsic growth capacity. We also conclude that myelin inhibitors and CSPGs are not the primary mechanism stopping axons at the DREZ.

## Introduction

The dorsal root (DR) carries primary sensory axons that project centrally from dorsal root ganglion (DRG) neurons to secondary neurons within the spinal cord and brainstem. DR injuries commonly result from brachial plexus, lumbosacral plexus, and cauda equina trauma, and may cause permanent loss of sensation, uncoordinated movement, and chronic pain (*Carlstedt, 2008*; *Kaiser et al., 2020*). The devastating consequences are because DR axons stop regenerating at the entrance of the spinal cord, the dorsal root entry zone (DREZ), and thus fail to restore connections with secondary neurons. Animal studies have reported functional recovery of nociception (*Ramer et al., 2000*; *Romero et al., 2001*; *Cafferty et al., 2007*; *Liu et al., 2009*; *Lin et al., 2014*; *Kelamangalath et al., 2015*), and, less frequently, of proprioception and mechanoreception (e.g., *Wang et al., 2008*; *Cheah et al., 2016*), for which large-diameter, myelinated proprio-/mechanoreceptive axons must regenerate far longer distances after crossing the DREZ.

Both neuron-intrinsic and -extrinsic inhibitors, which limit axon regrowth elsewhere in the injured CNS (*O'Shea et al., 2017*; *Griffin and Bradke, 2020*), are widely thought to block regeneration at the DREZ. Notably, however, unlike direct CNS injury, DR injury damages axons in the PNS without causing an impassable glial scar. Nevertheless, DR axons regenerating along the root quickly stop at

the scar-free DREZ, even after a nerve conditioning lesion (*Chong et al., 1999*; *Zhang et al., 2007*; *Di Maio et al., 2011*). This potent blockade is surprising because a nerve conditioning lesion sufficiently enhances the growth potential of a limited number of dorsal column (DC) axons to penetrate a glial scar after spinal cord injury (*Neumann and Woolf, 1999*; *Kwon et al., 2015*). Why the scar-free DREZ is impenetrable even to conditioning lesioned axons remains unclear, but myelin-associated inhibitors and extracellular matrix-associated chondroitin sulfate proteoglycans (CSPGs) are conventionally considered responsible (*Smith et al., 2012*; *Mar et al., 2016*). This view is based on reports that individually targeting myelin inhibitors or CSPGs produced robust regeneration of DR axons, including proprio-/mechanoreceptive axons, across the DREZ. Soluble peptides blocking interactions between myelin inhibitors and Nogo receptors were observed to dramatically enhance robust functional regeneration of myelinated, but not unmyelinated, axons after DR crush (*Harvey et al., 2009*; *Peng et al., 2010*). Similarly, blocking PTPσ, a CSPG receptor, was reported to produce functional regeneration of myelinated DR axons into the spinal cord (*Yao et al., 2019*). Activating integrins has been found to elicit long-distance, topographic and functional regeneration of both myelinated and unmyelinated DR axons, presumably by counteracting myelin inhibitors, CSPGs and tenascin-C (*Tan et al., 2011*; *Cheah et al., 2016*).

Incomplete lesions and conflicting results have also hampered conclusive agreement about the mechanism of growth inhibition at the DREZ, including about the primacy of myelin inhibitors and CSPGs. Although no published studies have contradicted the reports of robust regeneration after pharmacologically targeting myelin inhibitors, two groups have found that removing CSPGs alone enables only minimal penetration of DR axons through the DREZ (*Steinmetz et al., 2005*; *Wu et al., 2016*). CSPG removal, however, when combined with conditioning lesions, neurotrophic factors, or inflammation, has significantly enhanced intraspinal regeneration of DR axons (*Steinmetz et al., 2005*; *Wu et al., 2016*; *Guo et al., 2019*). Why eliminating CSPGs alone or a nerve conditioning lesion elicits only minimal regeneration across the DREZ is unknown, but the default assumption has been that myelin inhibitors alone are sufficiently potent to stop axons at the DREZ (*Smith et al., 2012*).

In the present work, we selectively traced regenerating proprio-/mechanoreceptive axons and used a novel wholemount assay to ensure that DR lesions were complete and the analysis comprehensive. Our analysis of triple knockout (tKO) mice lacking Nogo (A, B, C) (also known as Reticulon-4), MAG (myelin-associated glycoprotein), and OMgp/Omg (oligodendrocyte myelin glycoprotein), which is the first to genetically target all three major myelin inhibitors simultaneously, revealed that regeneration across the DREZ is not enhanced. Additionally, we found that supplemental removal of CSPGs in *Rtn4/Mag/Omg* tKO mice, the first combinatorial study to simultaneously eliminate myelin inhibitors and CSPGs, only modestly enhances regeneration of even conditioning lesioned DR axons. Thus, in contrast to the default assumption, which represents the prevalent view in the field, neither myelin inhibitors nor CSPGs, by themselves or even together, are sufficiently potent to prevent most DR axons from regenerating across the DREZ. Their absence, however, markedly and synergistically enhances intraspinal regeneration of glial cell line-derived neurotrophic factor (GDNF)-stimulated DR axons. These findings suggest the presence of inhibitory mechanism(s) of remarkably greater potency that potently blocks most axons at the DREZ, and that targeting myelin inhibitors and CSPGs can markedly enhance intraspinal penetration only when combined with an intervention that elevates axon growth capacity sufficiently robustly, above that achieved by a nerve conditioning lesion.

## Results

### Intraganglionic AAV2-GFP selectively labels proprioceptive and mechanoreceptive axons

Conventional assessment of DR regeneration has relied heavily on immunolabeling of tissue sections and consequently was subject to labeling artifacts and limited sensitivity. We initiated the present study by identifying a viral tracer that intensely and reliably labels regenerating DR axons. We tested various recombinant viral vectors carrying fluorescent reporters by microinjecting them into cervical DRGs of uninjured adult mice. Of those we examined at 2 weeks post-injection, AAV2-GFP (self-complementary adeno-associated virus serotype 2-enhanced green fluorescent protein) almost

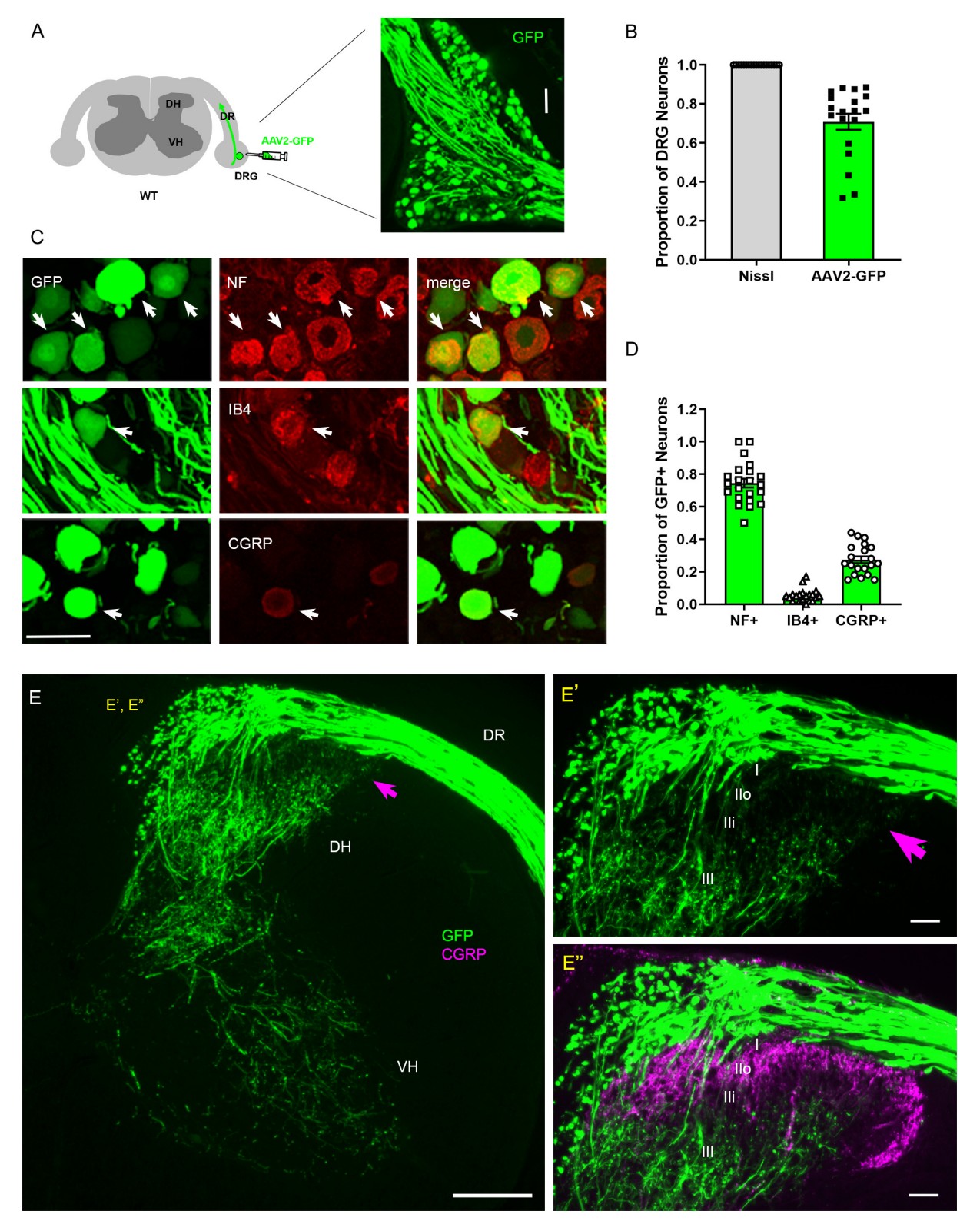

**Figure 1.** Intraganglionic AAV2-GFP labels proprioceptive and mechanoreceptive axons. (A) Schematic illustration of intraganglionic injection of scAAV2-eGFP and a representative dorsal root ganglion (DRG) showing infected neurons expressing GFP at 2 weeks post-injection. (B) Mice expressing GFP in >70% Nissl-stained neurons were used in the present study. (C) DRG transverse sections showing GFP+ neurons (arrows) co-expressing neurofilament (NF), IB4, or CGRP. (D) Quantitative comparisons of AAV2-GFP-infected neurons illustrating preferential labeling of large-diameter

*Figure 1 continued on next page*

*Figure 1 continued*

myelinated NF+ neurons, which mediate proprioception and mechanoreception. n > 20 sections, three mice. (**E**) A transverse section showing GFP+ axons along the root and within the right side of the spinal cord, projecting into dorsal column, deeper laminae of the dorsal horn and into the ventral horn. An arrow denotes superficial laminae I–IIi lacking GFP fluorescence. (**E', E"**) Enlarged views of the superficial dorsal horn, illustrating lack of GFP-fluorescence where CGRP+ nociceptive axons (magenta) innervate. DH: dorsal horn; DR: dorsal root; VH: ventral horn. Scale bars = 50 μm (**A, C, E', E"**), 200 μm (**E**).

exclusively transduced neurons, revealing brightly labeled cell bodies and axons (*Figure 1A*). After optimizing the virus titer, dosage, and microinjection technique, we were able to infect >70% neurons in most injections of DRGs (*Figure 1B*). Infected neurons included the three broadly classified subtypes of DRG neurons: large, neurofilament (NF)+ neurons, small- and medium-sized IB4+ non-peptidergic neurons and small CGRP+ peptidergic neurons (*Figure 1C*). Notably, a majority of the transduced, GFP-expressing neurons were NF+ (*Figure 1D*). In contrast, IB4+ neurons rarely were GFP+ and ~30% of GFP+ neurons were CGRP+ (*Figure 1D*), indicating that NF+ neurons were disproportionately transduced by AAV2-GFP. Consistent with the preferential infection of NF+ neurons, brightly labeled, large-diameter axons projected into the deeper layers of the dorsal horn (layer III–V) and into ventral horn, where large, myelinated axons terminate (*Figure 1E*). In contrast, superficial laminae of the dorsal horn, where small-diameter unmyelinated axons terminate (layer I, II), lacked GFP fluorescence (*Figure 1E', E"*), showing that AAV-GFP labels few if any IB4+ and CGRP+ axons. These findings are the first demonstration that AAV2-GFP predominantly transduces NF+ neurons and selectively reveals their proprio-/mechanoreceptive axons within the spinal cord.

NF+ neurons extend large-diameter myelinated axons that relay proprioception or mechanoreception via second-order neurons located deep in the spinal cord and in distant DC nuclei in the medulla (*Niu et al., 2013*). In contrast, IB4+ and CGRP+ neurons relay nociception through small-diameter unmyelinated axons that innervate secondary neurons in the superficial dorsal horn. Therefore, proprio-/mechanoreceptive axons require far more robust long-distance regeneration than nociceptive axons for functional recovery. Moreover, myelinated proprio-/mechanoreceptive axons regenerate more poorly than nonmyelinated nociceptive axons (*Tessler et al., 1988*; *Guseva and Chelyshev, 2006*; *Han et al., 2017*). Therefore, AAV2-GFP provides a unique opportunity to study selective regeneration of proprio-/mechanoreceptive axons whose regenerative capacity is particularly weak and needs robust augmentation.

## Strategies for complete lesions and comprehensive evaluation of DR regeneration

Regeneration studies in animals suffer from conflicting and non-reproducible results, in part due to incomplete lesions which lead to mistakenly interpreting spared axons as regenerating axons (*Steward et al., 2003*; *Steward et al., 2012*). Completely crushing a DR is particularly demanding because DRs are tightly attached to spinal cord surfaces in flat, transparent layers (*Han et al., 2012*; *Son, 2015*). Various surgical methods have been applied to facilitate complete lesions, such as repetitive and bidirectional crushing of a root (*Romero et al., 2001*; *Steinmetz et al., 2005*; *Wu et al., 2016*). However, there have been no assays that would confirm that a nerve crush surgery was successful.

We used two strategies to avoid spared axons. In one, we first crushed DRs and then microinjected AAV2-GFP into DRGs (*Figure 2A*). This strategy transduces only axons proximal to the lesion, leading to labeling of regenerating, but not degenerating, distal stump axons. This is important because distal axons are very slowly removed in the CNS (*Vargas and Barres, 2007*), and thus can be mistakenly identified as regenerating axons in a conventional immunostaining analysis of transverse sections. In the second strategy, after euthanizing a mouse typically at 2 weeks post injury (wpi), we harvested spinal cords with attached DRs and DRGs, examined them first in wholemounts, and excluded those with poor viral infections. We then carefully examined the properly labeled wholemounts and confirmed that lesions were complete (e.g., *Figure 2B*). We excluded those containing spared axons with the following characteristics: present in groups of only a few, relatively straight and extremely lengthy processes that extend along the entire length of the spinal cord and terminate with no discernible axon endings (e.g., *Figure 2C*; *Han et al., 2012*). A highly experienced

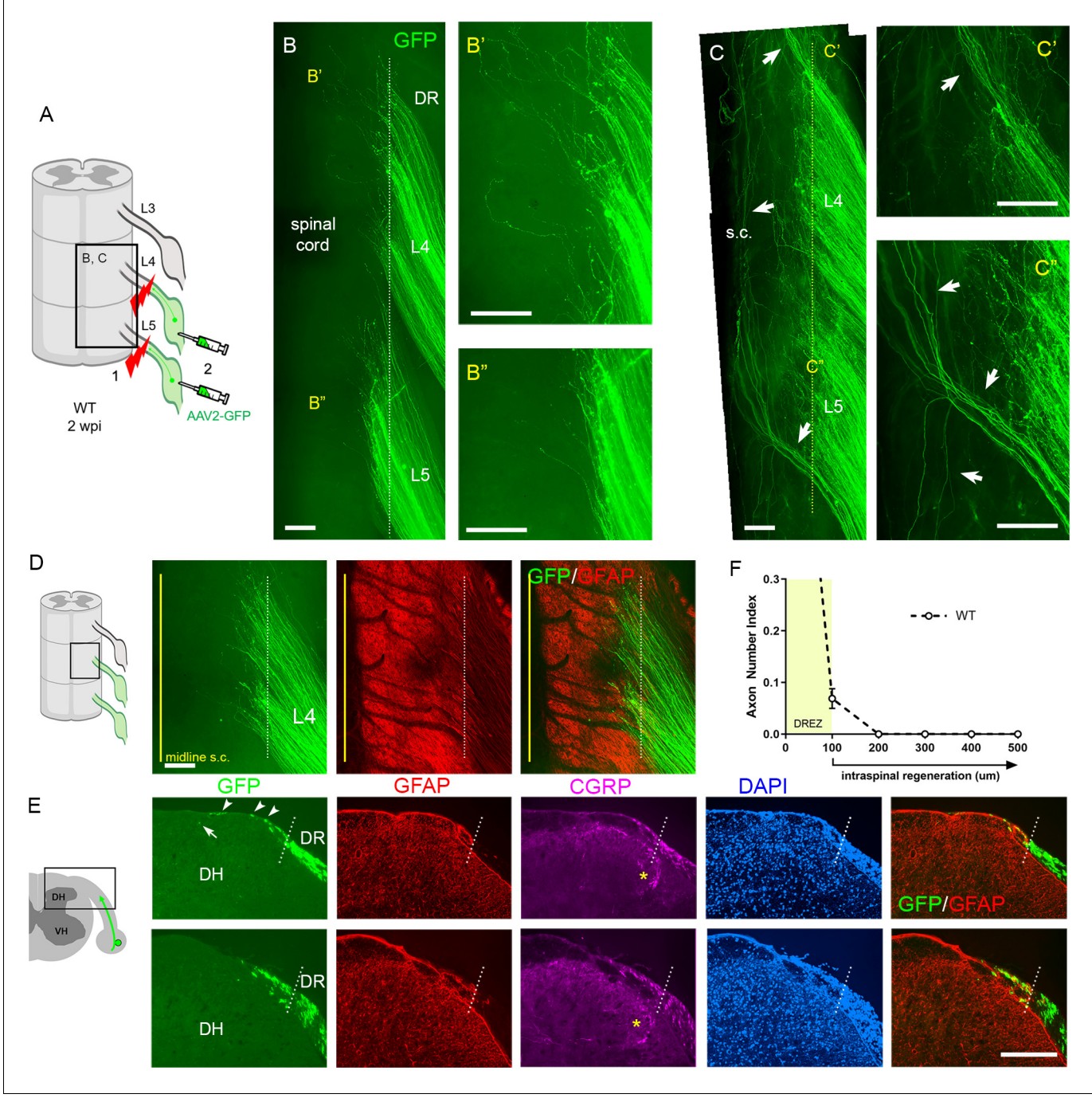

**Figure 2.** Additional strategies for complete lesions and evaluation of dorsal root (DR) regeneration. DR regeneration in wildtype (WT) mice assessed in wholemounts (A–D) and transverse sections (E, F) 2 weeks after L4 and L5 DR crush. (A) Schematic illustration of crushing roots prior to intraganglionic AAV2-GFP injections to avoid labeling of degenerating distal stump axons. (B) Wholemount view of completely crushed L4 and L5 DRs illustrating hundreds of GFP+ axons terminated at the entrance of spinal cord. (B′, B″) Enlarged views illustrating most axons terminated near the border. (C–C″) Wholemount views of incompletely crushed DRs showing spared axons with long intraspinal projections. Spared axons are easily detectable in wholemounts and commonly observed in the outermost dorsal rootlets (arrows). (D) Wholemount view of L4 dorsal root entry zone (DREZ) illustrating GFP+ axons that crossed the astrocyte: PNS border (dotted line) and terminated nearby. The astrocytic border is identified by GFAP immunostaining of astrocytes (red). Yellow line denotes spinal cord midline recognized by the midline vein. (E) Four-color immunolabeling of transverse sections illustrating limited penetration of GFP+ or CGRP+ axons through the DREZ. White dotted lines approximate the peripheral boundary of the DREZ (astrocyte: PNS border) by locating peripherally projecting astrocytic processes (red) or by greater abundance of cell nuclei in the PNS (blue). Axons rarely extended >200 μm beyond the border. Arrowheads denote frequently observed axons that grew along the growth-permissive dura. Arrow denotes occasionally observed subdural axons located several hundred microns past the border. (F) Quantitative analysis of DR regeneration on transverse

*Figure 2 continued on next page*

*Figure 2 continued*

sections (13 sections, three mice). ~90% GFP+ axons terminated within ~100 µm of the border. Axons growing farther than 100 µm are considered as having penetrated the DREZ. DH: dorsal horn; S.C.: spinal cord. Scale bars = 200 µm (**B–B″**, **C–C″**, **D**, **E**).

The online version of this article includes the following source data and figure supplement(s) for figure 2:

**Source data 1.** Source data for quantifying regeneration across the dorsal root entry zone.
**Figure supplement 1.** Side-by-side comparison of different mouse groups in wholemounts and transverse sections.
**Figure supplement 2.** Validation of DAPI as a marker of the CNS:PNS border.

surgeon performed all the root crushes. Nevertheless, incomplete lesions occurred in ~20% of the animals, typically because axons had been spared in the outermost dorsal rootlets (*Figure 2C*).

The wholemounts also enabled us to examine an unprecedented number of regenerating proprio-/mechanoreceptive axons from multiple injured roots. After a complete root crush in a wildtype (WT) mouse, hundreds of GFP+ axons all terminated at similar locations along the length of the dorsolateral spinal cord (*Figure 2B*, *Figure 2—figure supplement 1*). These GFP+ axons crossed the astrocyte:PNS border marked by GFAP (dotted lines) and terminated mostly within ~200 µm of the border, forming a narrow front of axon tips (*Figure 2D*). Following wholemount assessment, we prepared serial transverse sections and evaluated regeneration of DR axons, across the DREZ and within the spinal cord. In WT mice, axons frequently grew dorsally along the growth-permissive pia matter (*Figure 2E*, arrowheads). We occasionally observed axons located subdurally several hundred microns past the astrocyte:PNS border (*Figure 2E*, arrow). Most axons, however, were located within ~100 µm of the border and only a few axons reached ~200 µm (*Figure 2F*). In the present comparative analyses, we considered axons that grew farther than 100 µm from the border as having penetrated the DREZ (*Figure 2F*). When astrocytes could not be co-immunostained, the astrocyte:PNS border was identified by DAPI staining of cell nuclei that accumulate much more densely in the PNS than CNS. The borders delineated by GFAP and DAPI overlap closely with each other (*Figure 2E*). DAPI also delineate boundaries closely overlapped with those identified by laminin (*Figure 2—figure supplement 2*; see also *Figure 6—figure supplement 1*), another marker of the CNS:PNS border (*Ramer et al., 2004*; *Hoeber et al., 2017*), validating further our use of DAPI as an alternative boundary marker.

## Genetic deletion of Nogo, MAG, and OMgp elicits little regeneration across the DREZ

We first investigated the effects of simultaneous genetic deletion of myelin-associated inhibitors by examining global tKO mice lacking Nogo/Reticulon-4 isoforms (A, B, C), MAG, and OMgp. These mice were initially raised on a mixed background, extensively characterized, and used to study spinal cord regeneration (*Lee et al., 2010*). Our examination of these non-congenic tKO mutants revealed no enhanced regeneration of DR axons across the DREZ (data not shown). To overcome possible complications due to genetic background (*Montagutelli, 2000*; *Tedeschi et al., 2017*), we subsequently obtained *Rtn4/Omg* double KO and *Mag* KO mice raised on a C57BL/6 background and bred them to generate congenic *Rtn4/Mag/Omg* tKO mice (*Figure 3A*). Congenic tKO mice were viable and fertile, with no gross abnormalities. They were intercrossed to generate additional 2–3-month-old tKO mice; age-matched C57BL/6 mice were used as controls.

To examine DR regeneration, we unilaterally crushed L4 and L5 DRs ~3–5 mm from the DREZ and then microinjected high-titer AAV2-GFP into the ipsilateral L4 and L5 DRGs (*Figure 3B*). In this crush injury model, proximal axons of large, NF+ neurons are capable of regenerating across the injury site and growing along the root at ~1.5 mm/day until they are rapidly immobilized at the DREZ, ~4 days post injury (*Di Maio et al., 2011*). We examined WT and tKO at 2 wpi, which provides axons sufficient time to penetrate the DREZ if they are competent to do so. Wholemount examination of *Rtn4/Mag/Omg* tKO mice revealed many brightly labeled axons that extended along the L4 and L5 roots (*Figure 3D*), as they did also in WT mice (*Figure 2B*, *Figure 3C*). In both tKO and WT mice, however, most GFP+ axons terminated at similar longitudinal locations near the astrocyte:PNS border. Some axons extended substantially longer processes dorsally toward the spinal cord midline (*Figure 3D*, arrows). However, similar axons were also frequent in WT (*Figure 3C*, arrows), and their incidence and length were not noticeably different in tKO and WT mice. We next examined serial

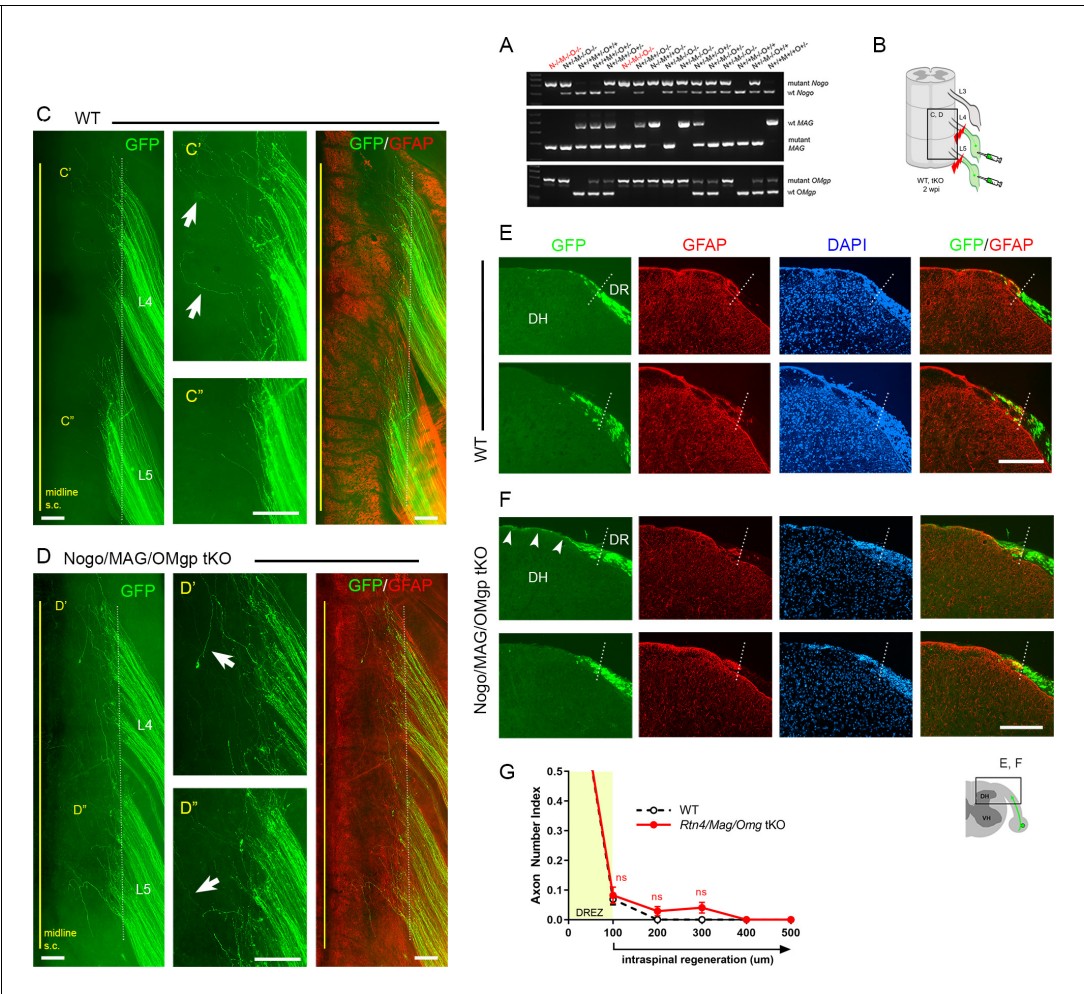

**Figure 3.** Genetic deletion of Nogo/MAG/OMgp elicits little intraspinal regeneration. Dorsal root (DR) regeneration in *Rtn4/Mag/Omg* triple knockout (tKO) mice assessed in wholemounts (**D**) or transverse sections (**F**) 2 weeks after L4 and L5 DR crush. (**A**) Identification of triple null mutants (red) lacking Nogo (**A, B, C**), MAG, and OMgp. (**B**) Schematic illustration of the experimental procedures. (**C**) Wholemount view of a wildtype (WT) mouse. (**C', C''**) Enlarged views of L4 and L5 dorsal root entry zone (DREZ) in (**C**). Arrows denote axons extending longer processes past the DREZ. (**D**) Wholemount views of a tKO mouse illustrating termination of hundreds of GFP+ axons near the astrocyte:PNS border (dotted line), as in WT mice. The astrocyte:PNS border is identified by GFAP immunostaining of astrocytes (red). (**D', D''**) Enlarged views of L4 and L5 DREZ in (**D**). Arrows denote axons extending longer processes past the DREZ, which were also frequently observed in WT mice. (**E**) Representative transverse sections of WT mice. (**F**) Representative transverse sections of *Rtn4/Mag/Omg* tKO mice illustrating little if any enhanced regeneration of GFP+ axons across the DREZ. Arrows denote axons that grew dorsally along the pia matter, as also observed in WT mice. (**G**) Quantitative comparisons illustrating no significant difference in WT and *Rtn4/Mag/Omg* tKO mice. 100 µm, p=0.9738, df = 162; 200 µm, p=0.5046, df = 162; 300 µm, p=0.1454, df = 162. Two-way repeated-measures ANOVA with Sidak's multiple comparisons test (WT: 13 sections, three mice; tKO: 16 sections, five mice). S.C.: spinal cord; ns: not significant. Scale bars = 200 µm (**C–C''**, **D–D''**, **E, F**).

The online version of this article includes the following source data for figure 3:

**Source data 1.** Source data for quantifying regeneration across the dorsal root entry zone.

transverse sections of L4 and L5 spinal cords prepared from tKO mice. No sections revealed GFP+ axons that crossed the DREZ and grew deep into the spinal cord. Most axons stopped at the DREZ, within ~100 µm of the astrocyte:PNS border, as in WT (*Figure 3E–G*). Axons that extended longer distances grew along the growth-permissive pia matter (*Figure 3F*, arrowheads), as in WT (*Figure 2E*, *Figure 3E*) Thus, in contrast to the earlier studies that reported robust regeneration of DR axons after pharmacological targeting of myelin signaling, genetic elimination of three major myelin inhibitors did not enable GFP+ axons to cross the DREZ. These results suggest that inhibiting myelin inhibitors alone is not sufficient to induce regeneration of proprio-/mechanoreceptive axons across the DREZ.

## Supplementary CSPG removal slightly increases regeneration across the DREZ in tKO mice

The limited regeneration across the DREZ in *Rtn4/Mag/Omg* tKO could be due to redundant inhibition by CSPGs that by themselves might be capable of arresting axons at the DREZ. Conversely, CSPG removal might induce only minimal regeneration (*Steinmetz et al., 2005*; *Wu et al., 2016*) due to redundant and potent inhibition by other inhibitors. To test this possibility, we attenuated CSPGs in *Rtn4/Mag/Omg* tKO using lentivirus encoding chondroitinase ABC (LV-chABC) (*Jin et al., 2011*). ChABC promotes axon regrowth by digesting growth-inhibitory glycosaminoglycan (GAG) chains on CSPGs (*Muir et al., 2019*). We unilaterally crushed L4 and L5 DRs of tKO mice, microinjected AAV2-GFP into the L4 and L5 DRGs, and injected high-titer LV-ChABC into the ipsilateral dorsal horn at multiple locations rostrocaudally along the L4–L5 DREZ (*Figure 4A*). Two weeks after injury, wholemounts of ChABC-expressed tKO mice appeared similar to those of WT mice: most GFP+ axons terminated near the astrocytic border (*Figure 4B, C*). We used CS56 antibody immunostaining to confirm that LV-ChABC effectively removed the inhibitory sulfated GAG chains on CSPGs (*Figure 4E*). Consistent with previous observations (*Han et al., 2017*), CSPG degradation was restricted to the dorsal horn on the injected side of the spinal cord (*Figure 4E*, asterisks). Notably, CSPGs were rapidly and markedly upregulated in Schwann cells after DR crush, resulting in far brighter CS-56 immunoreactivity in the PNS than in the CNS (data not shown). We observed considerable CS-56 immunoreactivity associated with Schwann cells near the DREZ in ChABC-expressed tKO mice (*Figure 4E'*, arrowheads). However, the intensity of immunoreactivity was markedly reduced compared to that in non-treated tKO mice, further suggesting that CSPGs at the DREZ were markedly attenuated by LV-ChABC.

Most of the serial transverse sections of L4 and L5 DREZ also showed DR axons arrested at the DREZ and were virtually indistinguishable from those of WT mice (*Figure 4D, E*). Some sections exhibited a few GFP+ axons located slightly deeper in the dorsal funiculus; such axons were not observed in WT or tKO (*Figure 4E'*, arrows). The number of axons at the DREZ was slightly increased in tKO compared to WT, as measured at 100 µm past the astrocyte:PNS border (*Figure 4F*), presumably reflecting locally enhanced axon outgrowth following degradation of endoneurial CSPGs at or near the DREZ (*Zuo et al., 1998*; *Graham and Muir, 2016*). Those axons that extended across the DREZ were within 200 µm of the astrocyte:PNS border and constituted only ~10% of GFP+ axons (*Figure 4F*), however, suggesting that additional attenuation of CSPGs in *Rtn4/Mag/Omg* tKO only modestly promoted regeneration across the DREZ. Thus, these findings, based on the first simultaneous targeting of myelin inhibitors and CSPGs, indicate that the limited regeneration in the absence of myelin inhibitors is unlikely because of CSPGs that by themselves might be capable of arresting most axons at the DREZ.

## Chronic regeneration failure at the DREZ despite absence of Nogo/MAG/OMgp and CSPGs

Concurrent ablation of myelin inhibitors and CSPGs only slightly enhanced regeneration, as assessed at 2 wpi, enabling only ~10% GFP+ axons to reach intraspinally ~100 µm past the DREZ. Additional axons may continue to penetrate the DREZ and grow within the spinal cord lacking myelin inhibitors and CSPGs. To investigate this possibility and the chronic effects of targeting myelin inhibitors and CSPGs, we next examined WT, *Rtn4/Mag/Omg* tKO, and ChABC-expressed tKO mice at 4 wpi (*Figure 5A*). Consistent with earlier studies of WT mice that demonstrated rapid and persistent immobilization of DR axons at the DREZ (*Golding et al., 1996*; *Di Maio et al., 2011*), we observed no enhanced regeneration across the DREZ at 4 wpi in WT mice, as examined in wholemounts (*Figure 5B*) or in transverse sections (*Figure 5C*). There was, however, a statistically insignificant increase in DR axons at the DREZ (*Figure 5D*).

Similarly, in *Rtn4/Mag/Omg* tKO at 4 wpi (*Figure 5E–G*), we found no qualitative or quantitative evidence that two additional weeks after injury enabled more axons to penetrate the DREZ. This finding suggests that DR axons were chronically immobilized as they entered the DREZ despite the absence of myelin inhibitors.

Likewise, ChABC-expressed *Rtn4/Mag/Omg* tKO mice exhibited little or no enhanced regeneration at 4 wpi, as examined in wholemounts (*Figure 5H*) and transverse sections (*Figure 5I*) (see also *Figure 2—figure supplement 1*). We observed no increase in GFP+ axons that crossed the DREZ,

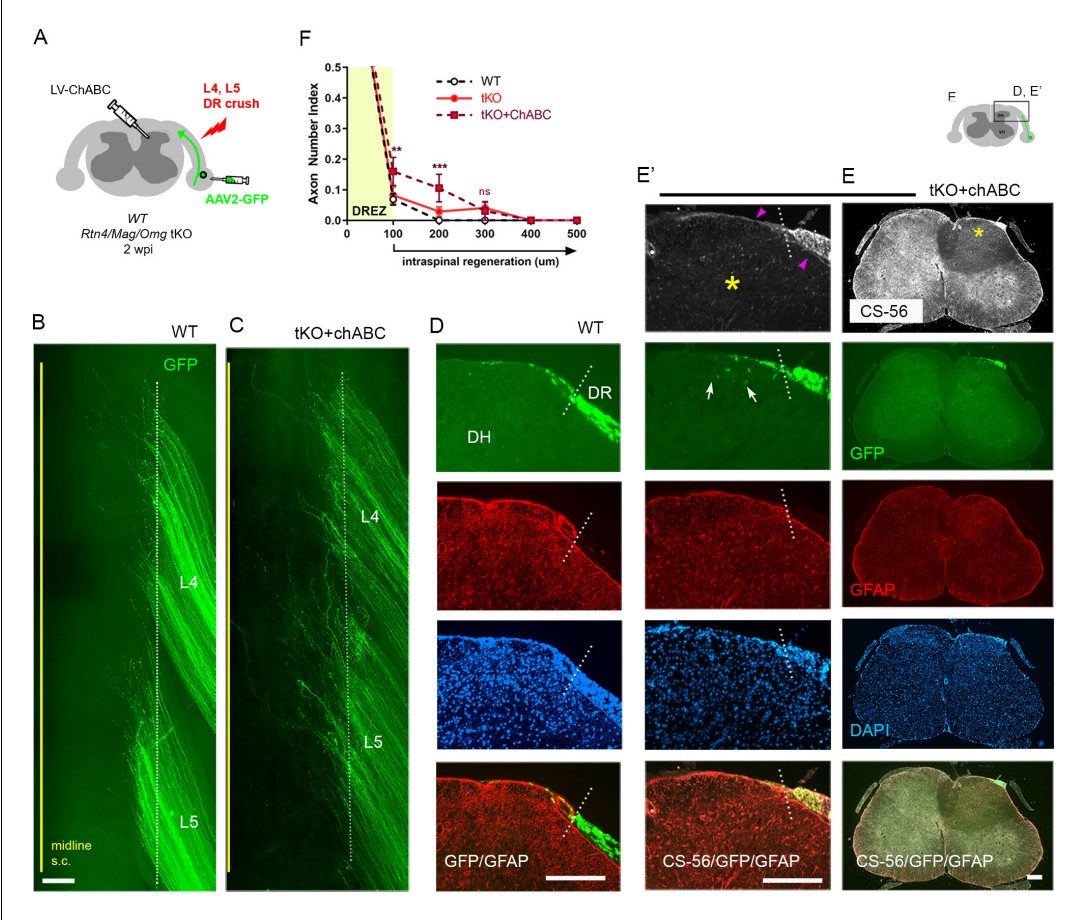

**Figure 4.** Additional chondroitin sulfate proteoglycan (CSPG) removal slightly increases intraspinal regeneration in triple knockout (tKO) mice. Dorsal root (DR) regeneration in chondroitinase ABC (ChABC)-expressed *Rtn4/Mag/Omg* tKO mice assessed in wholemounts (**C**) and transverse sections (**E–E'**) 2 weeks after L4 and L5 DR crush. (**A**) Schematic illustration of the experimental procedures. LV-ChABC was injected into ipsilateral dorsal horn at multiple locations rostrocaudally along the L4–L5 DREZ. (**B**) Wholemount views of a wildtype (WT) mouse. (**C**) Wholemount views of a ChABC-expressed tKO showing hundreds of GFP+ axons in L4 and L5 roots terminated near the astrocyte:PNS border (dotted line), as in WT and tKO mice. The astrocyte:PNS border is identified by GFAP immunostaining of astrocytes (red). (**D**) Representative transverse sections of a WT mouse. (**E**) Representative transverse sections of a ChABC-expressed tKO illustrating effective degradation of CSPGs and modestly enhanced intraspinal regeneration. CS-56 immunoreactivity is very low in ipsilateral dorsal horn (asterisks), indicating effective removal of inhibitory GAG chains of CSPGs. Arrowheads denote Schwann cell-associated CS-56 immunoreactivity, which is markedly reduced but discernible in ChABC-expressed tKO. (**E'**) Enlarged views showing a few GFP+ axons that penetrated the DREZ and are located at the top of the dorsal horn (arrows); such axons were not observed in WT or *Rtn4/Mag/Omg* tKO mice. (**F**) Quantitative comparisons illustrating modestly improved regeneration in ChABC-expressed *Rtn4/Mag/Omg* tKO mice: ~15% GFP+ penetrated the dorsal root entry zone (DREZ) and remained within ~200 μm of the border. ChABC-expressed tKO vs. WT: 100 μm, \*\*p=0.0022, df = 186; 200 μm, \*\*\*p=0.0003, df = 186; 300 μm, p=0.4818, df = 186. ChABC-expressed tKO vs. tKO: 100 μm, \*\*p=0.0086, df = 186; 200 μm, \*\*p=0.0099, df = 186; 300 μm, p=0.9262, df = 186. Two-way repeated-measures ANOVA with Sidak's multiple comparisons test (WT: 13 sections, three mice; tKO: 16 sections, five mice; ChABC-tKO: 14 sections, three mice). S.C.: spinal cord; ns: not significant. Scale bars = 200 μm (**B, C, D, E–E'**).

The online version of this article includes the following source data for figure 4:

**Source data 1.** Source data for quantifying regeneration across the dorsal root entry zone.

indicating that no additional axons penetrated the DREZ and that those axons intraspinally located at 2 wpi (~10% GFP+ axons) did not extend further within the spinal cord lacking CSPGs. We observed significantly more axons at the DREZ in ChABC-expressed *Rtn4/Mag/Omg* tKO mice at 4 wpi than at 2 wpi (*Figure 5J*) or than in WT or *Rtn4/Mag/Omg* tKO mice at 4 wpi (*Figure 5K*). Notably, however, these GFP+ axons remained at the DREZ, indicating that prolonged attenuation of CSPGs did not enable them to penetrate the DREZ lacking myelin inhibitors.

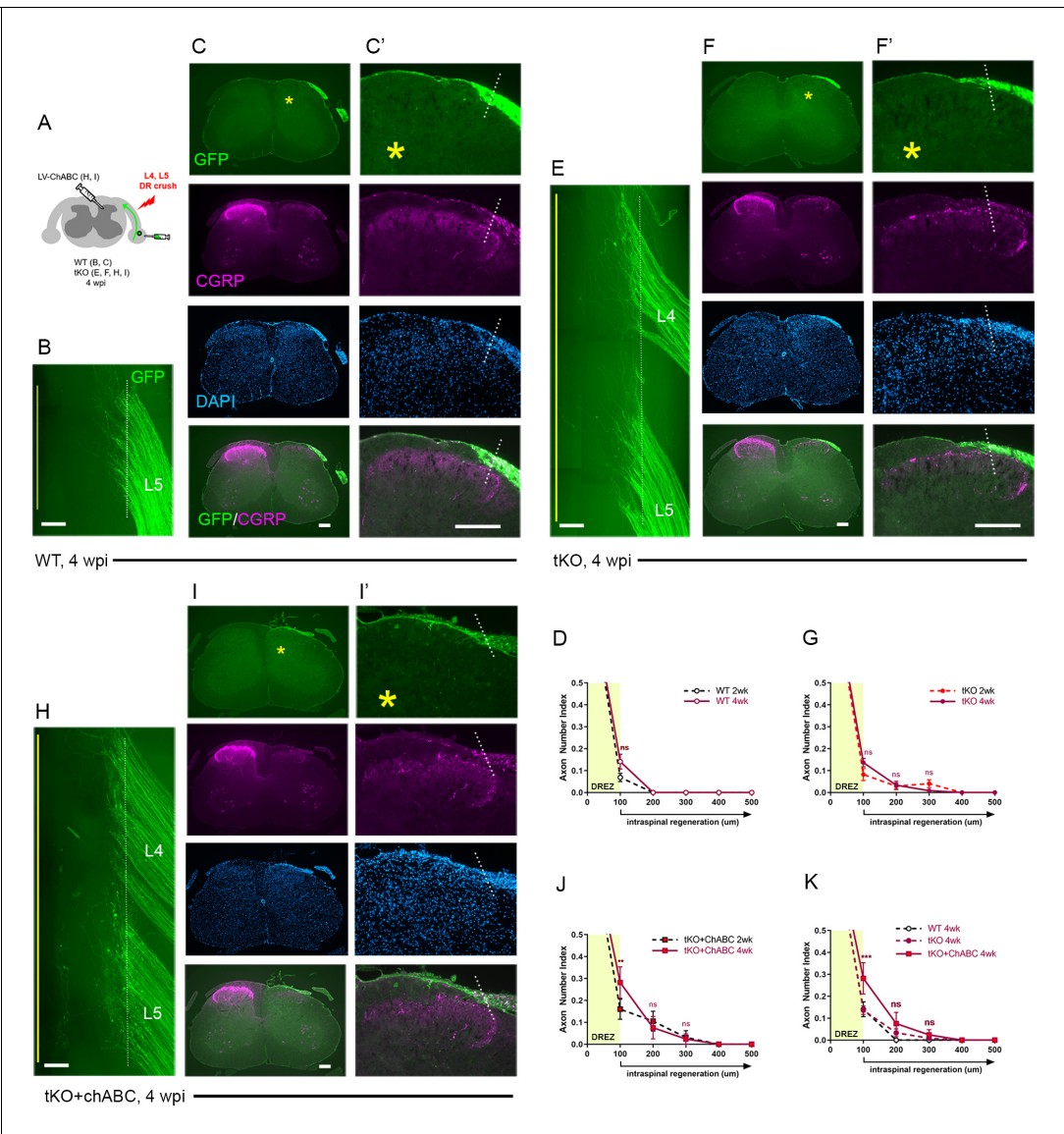

**Figure 5.** Chronic regeneration failure at the dorsal root entry zone (DREZ) lacking Nogo/MAG/OMgp and chondroitin sulfate proteoglycans (CSPGs). Dorsal root (DR) regeneration in wildtype (WT) (A–D), *Rtn4/Mag/Omg* triple knockout (tKO) (E–G), and chondroitinase ABC (ChABC)-expressed *Rtn4/Mag/Omg* tKO mice (H–K) analyzed 4 weeks after L4 and L5 DR crush. (A) Schematic illustration of the experimental procedures. (B) Wholemount view of L5 DREZ in a WT mouse showing no noticeably enhanced regeneration into the spinal cord. (C, C') Transverse sections showing no improved penetration of GFP+ (green) and CGRP+ axons (magenta) through the DREZ. (D) Quantitative comparisons of WT mice at 2 weeks post injury (wpi) and 4 wpi illustrating no significant difference. 100 μm, p=0.5292, df = 102. Two-way repeated-measured ANOVA with Sidak's multiple comparisons test (WT-2 wpi: 13 sections, three mice; WT-4 wpi: 14 sections, four mice). (E) Wholemount view of L4–L5 DREZ in a tKO showing no marked increase in intraspinal regeneration. (F, F') Transverse sections of a tKO mouse showing GFP+ (green) and CGRP+ axons (magenta) remaining at the DREZ at 4 wpi. (G) Quantitative comparisons of tKO mice at 2 wpi and 4 wpi illustrating no significant difference. 100 μm, p=0.5067, df = 168; 200 μm, p>0.9999, df = 168; 300 μm, p>0.9999, df = 168. Two-way repeated-measures ANOVA with Sidak's multiple comparisons test (tKO-2 wpi: 16 sections, five mice; tKO-4 wpi: 14 sections, five mice). (H) Wholemount view of L4–L5 DREZ in a ChABC-expressed tKO showing no noticeably enhanced intraspinal regeneration. (I, I') Transverse sections of a ChABC-expressed tKO showing GFP+ (green) and CGRP+ axons (magenta) remaining at the DREZ at 4 wpi. (J) Quantitative comparisons of ChABC-expressed tKO mice at 2 wpi and 4 wpi illustrating no significant increase in GFP+ axons that penetrated the DREZ. 100 μm, p=0.0027, df = 60; 200 μm, p=0.936, df = 60; 300 μm, p>0.9999, df = 60. Two-way repeated-measures ANOVA with Sidak's multiple comparisons test (ChABC-tKO-2 wpi: 14 sections, three mice; ChABC-tKO-4 wpi: 12 sections, three mice). (K) Quantitative comparisons of WT, tKO, and ChABC-expressed tKO at 4 wpi showing no significant difference in GFP+ axons crossing the DREZ. ChABC-expressed tKO vs. WT: 100 μm, ***p=0.0001, df = 144; 200 μm, p=0.668, df = 144; 300 μm, p=0.7582, df = 144. Two-way repeated-measures ANOVA with Sidak's multiple comparisons test. Scale bars = 200 μm (B, C–C', E–F', H–I').

The online version of this article includes the following source data for figure 5:

**Source data 1.** Source data for quantifying regeneration across the dorsal root entry zone.

We also examined regeneration of CGRP+ nociceptive axons, which are not discernibly labeled by AAV2-GFP. As anticipated, it was often not possible, as noted in WT mice at 2 wpi, to evaluate intraspinal regeneration on sections due to punctate residual CGRP immunoreactivity associated with superficial laminae of the ipsilateral dorsal horn, particularly in Lissauer's tract (*Figure 2E*, asterisks). Dorsal funiculus above the dorsal horn, however, largely lacked CGRP+ immunoreactivity, indicating little if any penetration of CGRP+ axons through the DREZ at 2 wpi in WT mice (*Figure 2E*). Similarly, dorsal funiculus lacked CGRP immunoreactivity at 4 wpi in WT (*Figure 5C*), *Rtn4/Mag/Omg* tKO (*Figure 5F*), and ChABC-expressed *Rtn4/Mag/Omg* tKO mice (*Figure 5I*). CGRP immunoreactivity in superficial laminae of these mice at 4 wpi appeared similar to that in WT mice at 2 wpi. Moreover, it was not directly connected to CGRP+ axons at the DREZ on any sections, indicating that no axons reached superficial laminae directly through the DREZ. Thus, CGRP+ axons also stop at the DREZ even with concurrent prolonged absence of myelin inhibitors and CSPGs.

Collectively, these studies, which represent the first evaluation of DR regeneration in the prolonged and concurrent absence of myelin inhibitors and CSPGs, demonstrate that both myelinated and unmyelinated DR axons stop at the DREZ and continue to be immobilized, even if myelin inhibitors and CSPGs are simultaneously removed. Therefore, the minimal regeneration observed in earlier studies targeting CSPGs alone was not due to redundant inhibition by myelin inhibitors. Instead, our findings suggest that neither myelin inhibitors nor CSPGs, by themselves or even together, are sufficiently potent to prevent regeneration of most DR axons across the DREZ.

## DR axons fail to penetrate the DREZ in *Rtn4/Mag/Omg* tKO even after a nerve conditioning lesion

Although genetic deletion of myelin inhibitors alone does not permit DR axons to grow through the DREZ, it may enable conditioned axons with enhanced growth capacity to do so. To test this possibility, we performed a single nerve crush conditioning lesion that alone fails to enhance regeneration across the DREZ (*Chong et al., 1999*; *Zhang et al., 2007*; *Di Maio et al., 2011*) by crushing the sciatic nerve 10 days before crushing the L4 and L5 roots (*Figure 6A*). At 2 wpi, wholemounts of GFP+ axons in conditioned *Rtn4/Mag/Omg* tKO mice did not show noticeably enhanced penetration of the DREZ (*Figure 6C*). Accordingly, serial transverse sections occasionally exhibited a few axons that reached dorsolateral gray matter (*Figure 6E*, arrow), but most GFP+ axons failed to penetrate the DREZ (*Figure 6E, F*). Dorsal funiculus lacked CGRP immunoreactivity, suggesting that conditioning lesioned CGRP+ axons also failed to penetrate the DREZ. Compared to WT and tKO mice (*Figure 6F*, see also *Figure 2—figure supplement 1*), conditioned tKO mice exhibited significantly more axons at the DREZ, presumably reflecting enhanced regeneration in the peripheral portion of the DR (*Di Maio et al., 2011*). We rarely observed axons extended more than >200 µm from the astrocyte:PNS border in conditioned tKO mice, as in WT and non-conditioned tKO mice (*Figure 6F*). Therefore, removal of myelin inhibitors does not enable nerve crush conditioned axons to regenerate beyond the DREZ.

## Double conditioning lesion modestly enhances regeneration of DR axons across the DREZ in WT mice

A single nerve crush does not provide maximal mechanical conditioning of DRG neurons. *Neumann et al., 2005* observed that double conditioning lesions dramatically enhance regeneration of DC axons after spinal cord injury, presumably because the second nerve injury at 1 wpi sustains the enhanced growth capacity of DRG neurons. Because the effects of double conditioning lesions on DR regeneration have not been studied, we investigated whether regeneration across the DREZ is enhanced. Following the double conditioning paradigm of Neumann et al., we transected the ipsilateral sciatic nerve 3 days before and 7 days after L4 and L5 DR crush in WT mice (*Figure 6—figure supplement 1A*). At 2 wpi, wholemounts of GFP+ axons in double lesioned WT mice did not show enhanced penetration of the DREZ (*Figure 6—figure supplement 1D*) compared to WT or tKO mice conditioned by a single nerve crush (*Figure 6—figure supplement 1B, C*). Notably, however, serial transverse sections frequently exhibited axons that entered dorsolateral gray matter (*Figure 6—figure supplement 1G, H*, arrows). Nonetheless, most GFP+ axons (>90%) failed to penetrate the DREZ and only a few reached 200 µm beyond the DREZ (*Figure 6—figure supplement 1J*). Quantitative comparisons indicate that the double conditioning lesion enhanced regeneration in

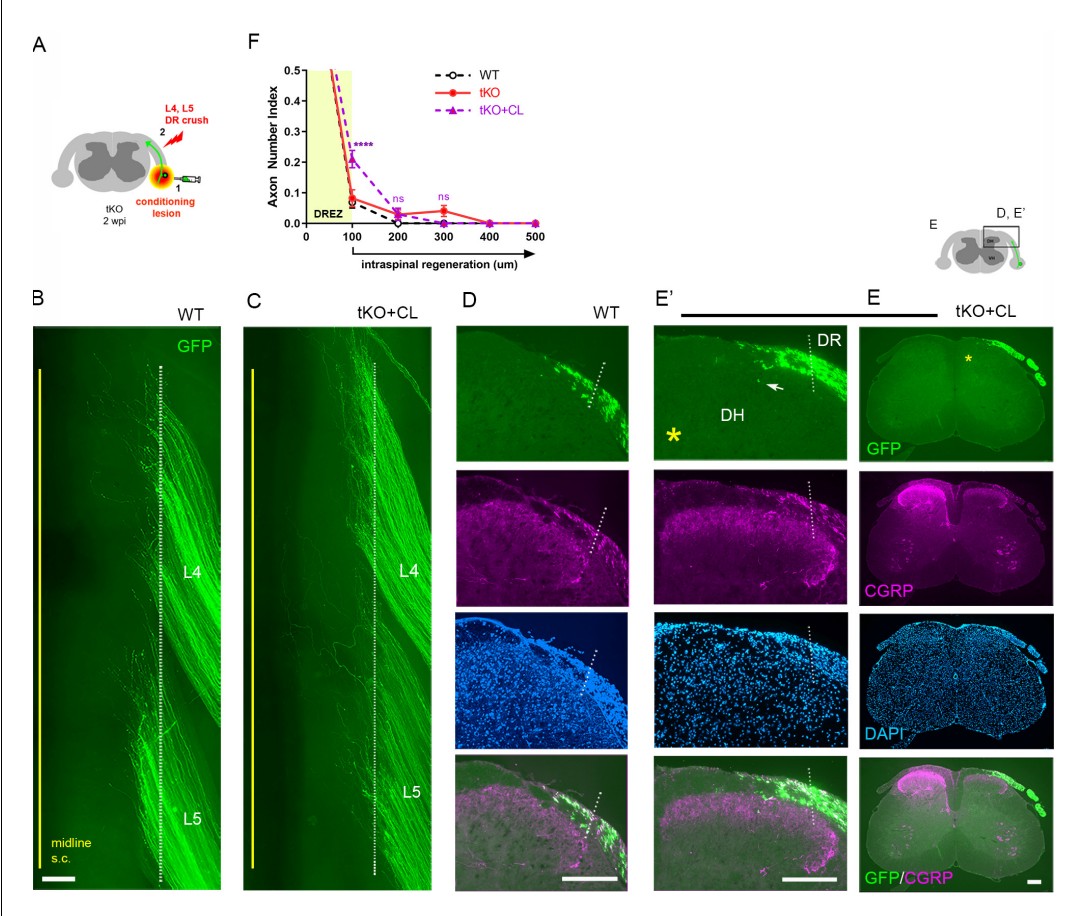

**Figure 6.** Nerve conditioning lesion does not promote regeneration across the dorsal root entry zone (DREZ) in triple knockout (tKO) mice. (A) Schematic illustration of the experimental procedures. *Rtn4/Mag/Omg* tKO mice received a nerve conditioning lesion 10 days before L4 and L5 dorsal root (DR) crush and were assessed at 2 weeks post injury (wpi). (B) Wholemount view of a wildtype (WT) mouse. (C) Wholemount view of a conditioned tKO showing hundreds of GFP+ axons terminated near the astrocyte:PNS border (dotted line), as in WT and tKO mice. (D) Transverse sections of a WT mouse. (E, E') Transverse sections of a conditioned tKO illustrating little if any enhanced regeneration of GFP+ (green) or CGRP+ axons (magenta) across the DREZ. An arrow denotes occasionally observed GFP+ axons that reached dorsolateral gray matter. (F) Quantitative comparisons illustrating no significant difference in WT, tKO, and conditioned tKO mice. tKO vs. conditioned tKO: 100 μm, ****p<0.0001, df = 220; 200 μm, p=0.9991, df = 220; 300 μm, p>0.9999, df = 220. Two-way repeated-measures ANOVA with Sidak's multiple comparisons test (WT: 13 sections, three mice; tKO: 16 sections, five mice; conditioned-tKO: 11 sections, three mice). ns: not significant. Scale bars = 200 μm (B, C, D, E, E').

The online version of this article includes the following source data and figure supplement(s) for figure 6:

**Source data 1.** Source data for quantifying regeneration across the dorsal root entry zone.
**Figure supplement 1.** Double conditioning lesion modestly enhances regeneration across the dorsal root entry zone (DREZ) in wildtype (WT) mice.
**Figure supplement 1—source data 1.** Source data for quantifying regeneration across the dorsal root entry zone.

WT mice to a level comparable to or slightly greater than a single crush conditioning lesion in *Rtn4/Mag/Omg* tKO mice (*Figure 6—figure supplement 1J*). However, it was far less efficacious than intraspinally expressed GDNF, which enabled many more GFP+ axons (>40%) to cross the DREZ and extend deeper in the spinal cord (*Figure 6—figure supplement 1I, J*; see also Figure 8D). Thus, like a single nerve crush conditioning lesion with supplementary removal of myelin inhibitors, a double conditioning lesion, although reported to markedly enhance regeneration of DC axons, failed to enable most DR axons to cross the DREZ after DR injury.

## Supplementary CSPG removal minimally enhances regeneration of conditioning lesioned axons across the DREZ in tKO mice

We next asked whether additional attenuation of CSPGs in *Rtn4/Mag/Omg* tKO mice would enable robust penetration of conditioned axons through the DREZ. To test this possibility, we injected LV-ChABC rostrocaudally in dorsal horns along the L4–L5 DREZ in *Rtn4/Mag/Omg* tKO mice that received a single nerve crush conditioning lesion 10 days before crushing L4 and L5 roots (*Figure 7A*). Surprisingly, at 2 wpi, despite the concurrent degradation of CSPGs, conditioned axons of *Rtn4/Mag/Omg* tKO mice largely remained near the astrocyte:PNS border in wholemounts (*Figure 7C*, *Figure 2—figure supplement 1*). CS-56 antibody immunostaining confirmed effective removal of CSPGs in dorsal horns (*Figure 7E*, asterisk). Serial transverse sections occasionally revealed GFP+ axons that reached dorsolateral gray matter (*Figure 7F*, arrows), but most axons remained at the DREZ within ~100 μm of the border (*Figure 7G*). Similarly, we observed no noticeably enhanced regeneration of CGRP+ axons in conditioned/ChABC-expressed *Rtn4/Mag/Omg* tKO (*Figure 7F*). These results demonstrate that additional removal of CSPGs did not enable significantly more GFP+ axons to penetrate the DREZ in conditioned tKO mice. In sum, CSPG degradation and/or a nerve conditioning lesion increases the number of axons at the DREZ in *Rtn4/Mag/Omg* tKO, but most axons (>90%) fail to extend across the DREZ (*Figure 7H*). CSPG removal improves penetration of conditioned or unconditioned axons in tKO mutant, but the effect is modest, enabling only ~10% axons to extend ~100 μm past the DREZ (*Figure 7H*). These findings collectively suggest that the inhibitory role of myelin inhibitors and CSPGs at the DREZ, by themselves or even together, is moderate: they are not the primary factors that arrest even conditioning lesioned DR axons at the DREZ.

## Nogo/MAG/OMgp removal markedly enhances intraspinal regeneration of GDNF-stimulated DR axons

Our findings that conditioning lesioned axons penetrate only modestly through the DREZ in ChABC-expressed tKO mice suggest that myelin inhibitors and CSPGs do not strongly inhibit DR axons. We also found that a double conditioning lesion fails to elicit robust DR regeneration, indicating that mechanical conditioning does not sufficiently enhance the intrinsic growth capacity of DR axons to cross the DREZ. If myelin inhibitors and CSPGs indeed play only a moderate inhibitory role at the DREZ, greater enhancement of intrinsic growth capacity may be essential for robust penetration in their absence. In addition, myelin inhibitors and CSPGs may hinder regeneration within the spinal cord, although their inhibition is insufficient to block most axons at the DREZ. To test further the efficacy of myelin inhibitors and CSPGs, we applied a treatment that enabled many axons to penetrate the DREZ and then examined if their intraspinal regeneration was altered in *Rtn4/Mag/Omg* tKO mice with or without additional attenuation of CSPGs. Intraspinal expression of neurotrophic factors elevates regenerative capacity and chemotropically attracts DR axons, enabling them to cross the DREZ (*Ramer et al., 2000*; *Iwakawa et al., 2001*; *Kelamangalath et al., 2015*). When expressed virally within the spinal cord of WT mice, GDNF promoted penetration of many large-diameter DR axons through the DREZ and further into the spinal cord (*Kelamangalath et al., 2015*; *Han et al., 2017*).

We microinjected lentivirus expressing GDNF (LV-GDNF), extensively used in earlier studies (*Deng et al., 2013*; *Zhang et al., 2013*; *Kelamangalath et al., 2015*; *Han et al., 2017*), into the dorsal horn rostrocaudally along the L4–L5 DREZ at the time of DR crush in WT (*Figure 8A*) or *Rtn4/Mag/Omg* tKO mice (*Figure 8E*). At 2 wpi, wholemounts of GDNF-expressed WT mice did not show markedly enhanced regeneration into the spinal cord (*Figure 8B*). Serial sections, however, frequently revealed GFP+ axons that crossed the DREZ and extended further into the dorsal funiculus and gray matter (*Figure 8C*). Quantification indicated that ~50% of GFP+ axons penetrated the DREZ and extended up to ~300 μm from the astrocyte:PNS border (*Figure 8D*). Consistent with an earlier study (*Iwakawa et al., 2001*), we also observed numerous CGRP+ axons in the dorsal funiculus, indicating regeneration of nociceptive axons across the DREZ in GDNF-expressed WT mice (*Figure 8C*).

Compared to GDNF-expressed WT mice, wholemounts of GDNF-expressed *Rtn4/Mag/Omg* tKO mice more frequently revealed areas of the DREZ that exhibited conspicuously intense GFP fluorescenc, due to densely accumulated subdural GFP+ axons (*Figure 8F*, arrows; see also *Figure 2—*

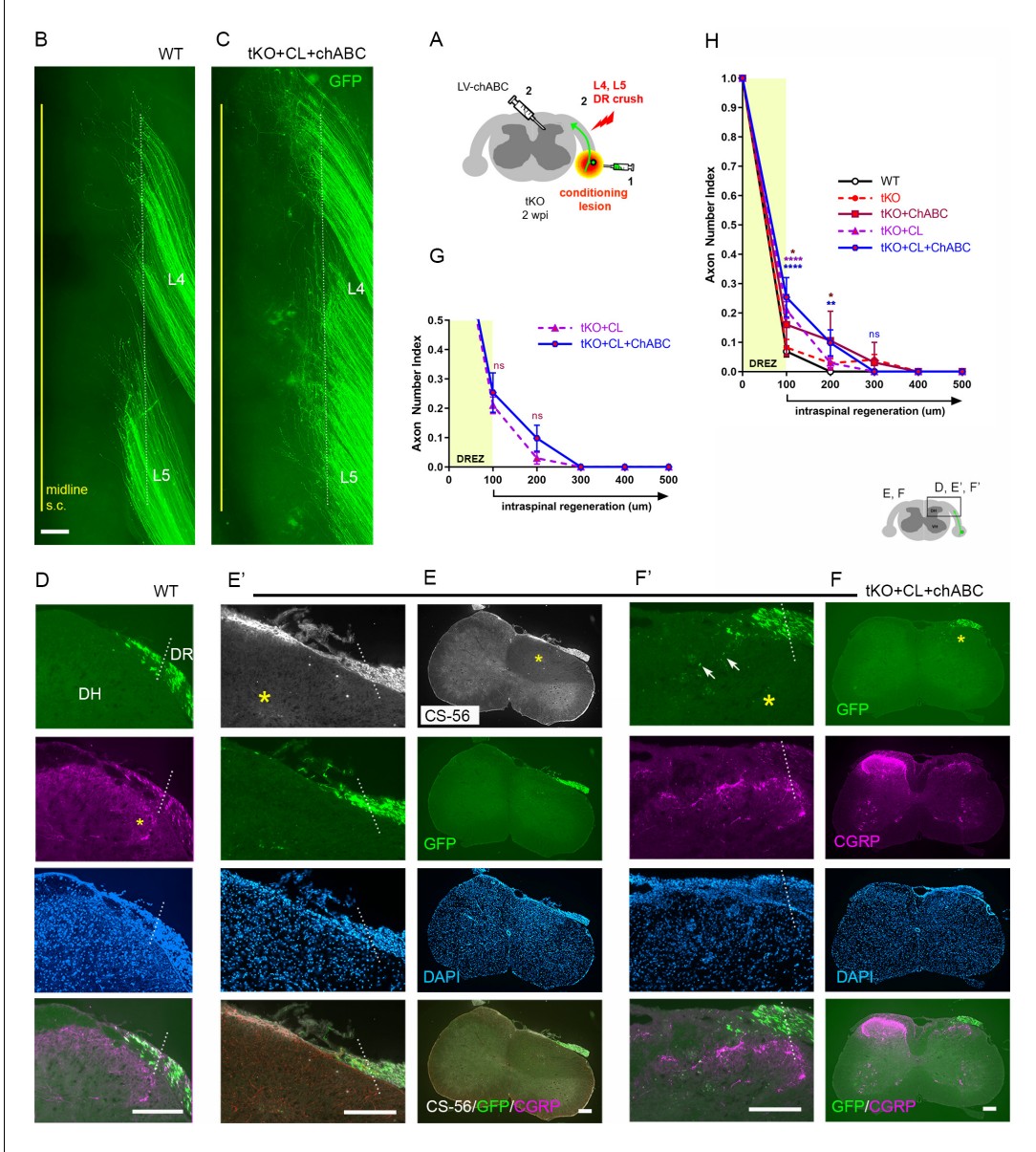

**Figure 7.** Additional chondroitin sulfate proteoglycan (CSPG) removal minimally enhances regeneration of conditioning lesioned axons in triple knockout (tKO) mice. (A) Schematic illustration of the experimental procedures. LV-chondroitinase ABC (LV-ChABC) was injected into *Rtn4/Mag/Omg* tKO mice that received a conditioning lesion 10 days before L4 and L5 dorsal root (DR) crush. (B) Wholemount view of a wildtype (WT) mouse. (C) Wholemount view of a ChABC/conditioned tKO showing hundreds of GFP+ axons that remain near the border (dotted line). (D) Transverse sections of a WT mouse. (E–E') Transverse sections of a ChABC/conditioned tKO illustrating effective CSPG degradation confirmed by the lack of CS-56 immunoreactivity (asterisks) and little if any intraspinal regeneration of GFP+ axons. (F–F') Additional transverse sections of a ChABC/conditioned tKO illustrating limited intraspinal regeneration of GFP+ or CGRP+ axons (magenta). Arrows denote occasionally observed GFP+ axons that enter dorsal gray matter. (G) Quantitative comparisons illustrating no significant difference in ChABC/conditioned tKO and conditioned tKO mice. 100 μm, p=0.7629, df = 114; 200 μm, p=0. 2671, df = 114. Two-way repeated-measures ANOVA with Sidak's multiple comparisons test (conditioned tKO: 11 sections, three mice; ChABC/conditioned tKO: 10 sections, four mice). (H) Quantitative summary illustrating minimal intraspinal regeneration of even conditioned axons after concurrent removal of myelin inhibitors and CSPGs; only ~10% GFP+ axons extended ~100 μm past the dorsal root entry zone (DREZ). 100 μm, *p=0.0488, df = 300 (WT vs. ChABC expressed tKO), ****p<0.0001, df = 300 (WT vs. conditioned tKO, WT vs. ChABC/conditioned tKO); 200 μm, *p=0.014, df = 300 (WT vs. ChABC expressed tKO), **p=0.0024, df = 300 (WT vs. ChABC/conditioned tKO); 300 μm, p>0.9999; df = 300. Two-way repeated-measures ANOVA with Sidak's multiple comparisons test. Scale bars = 200 μm (B, C, D, E–E', F–F').
The online version of this article includes the following source data for figure 7:

**Source data 1.** Source data for quantifying regeneration across the dorsal root entry zone.

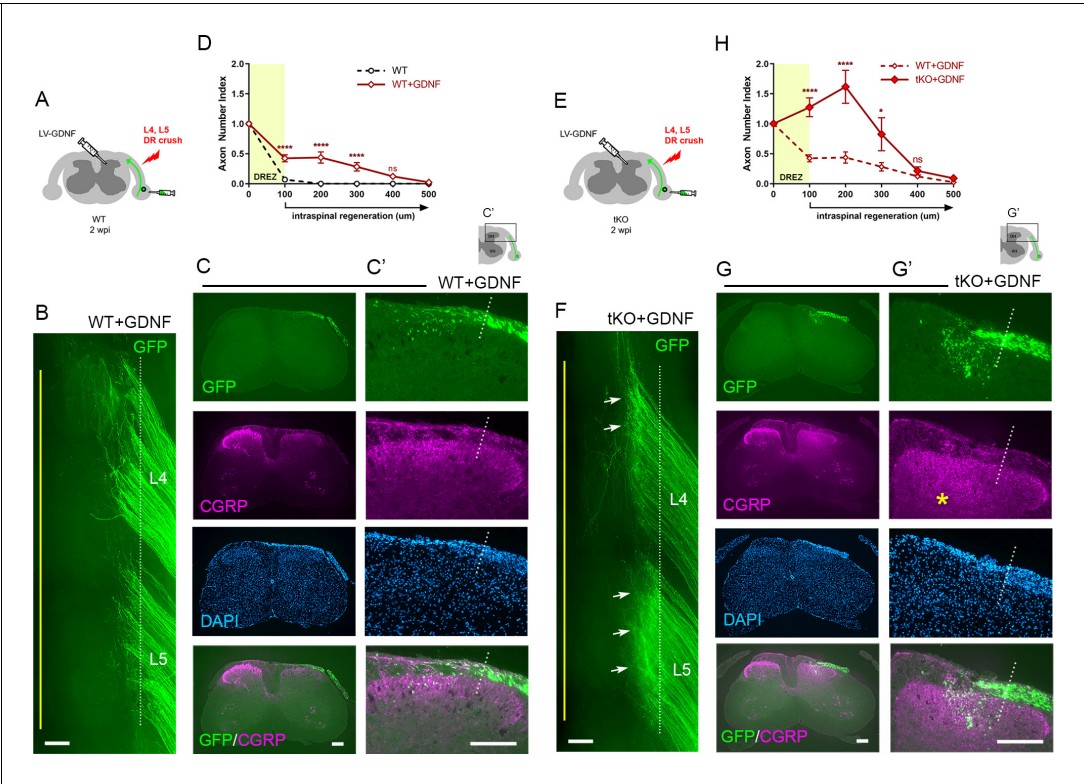

**Figure 8.** Nogo/MAG/OMgp removal markedly enhances regeneration of glial cell line-derived neurotrophic factor (GDNF)-stimulated dorsal root (DR) axons. GDNF-induced intraspinal regeneration analyzed in wildtype (WT) (**A–D**) and *Rtn4/Mag/Omg* triple knockout (tKO) mice (**E–H**) 2 weeks after L4 and L5 DR crush. (**A**) Schematic illustration showing intraspinal injections of LV-GDNF at the time of root crush and AAV2-GFP injections in WT mice. (**B**) Wholemount view of a GDNF-expressed WT illustrating hundreds of GFP+ axons largely remaining near the border. (**C–C'**) Transverse sections of a GDNF-expressed WT showing a number of GFP+ (green) or CGRP+ axons (magenta) that cross the DREZ and extend further into the dorsal funiculus and gray matter. (**D**) Quantitative comparisons illustrating significantly enhanced penetration of GFP+ axons across the DREZ. 100 µm, 200 µm, and 300 µm, ****p<0.0001, df = 156; 400 µm, p=0.2446, df = 156. Two-way repeated-measures ANOVA with Sidak's multiple comparisons test (WT: 13 sections, three mice; GDNF-expressed WT: 15 sections, three mice). (**E**) Schematic illustration of the experimental procedures in *Rtn4/Mag/Omg* tKO mice. (**F**) Wholemount view of a GDNF-expressed tKO mouse revealing intensely fluorescent area of the L4 and L5 DREZ (arrows), likely due to densely accumulated subdural GFP+ axons. (**G–G'**) Transverse sections of a GDNF-expressed tKO mouse displaying numerous GFP+ axons regenerating deep into dorsal horn. Asterisks denote CGRP immunoreactivity in deep dorsal laminae (magenta), presumably indicating enhanced regeneration of CGRP+ axons in GDNF-expressed tKO, as compared to that in GDNF-expressed WT. (**H**) Quantitative comparisons illustrating markedly greater intraspinal growth of GFAP-labeled axons in GDNF-expressed tKO than in GDNF-expressed WT mice. 100 µm and 200 µm, ****p<0.0001, df = 174; 300 µm, *p=0.028, df = 174; 400 µm, p=0.9975, df = 174. Two-way repeated-measures ANOVA with Sidak's multiple comparisons test (GDNF-expressed WT: 15 sections, three mice; GDNF-expressed tKO: 16 sections, five mice). ns: not significant. Scale bars = 200 µm (**B, C, C', F, G, G'**).

The online version of this article includes the following source data for figure 8:

**Source data 1.** Source data for quantifying regeneration across the dorsal root entry zone.

*figure supplement 1*). Indeed, serial sections often displayed abundant GFP+ axons at and beyond the DREZ, indicating markedly enhanced regeneration of GDNF-stimulated axons across the DREZ in tKO mice (*Figure 8G*). We also observed bright CGRP+ immunoreactivity in deep dorsal laminae that are usually devoid of residual CGRP+ staining, presumably indicating greatly enhanced regeneration of nociceptive axons in tKO mice (*Figure 8G*, asterisk). The number of axons that crossed the DREZ far exceeded that observed in GDNF-expressed WT mice, nearly tripling the number of GFP+ axons at 200 µm from the astrocyte:PNS border (*Figure 8H*). Therefore, elimination of myelin inhibitors enabled many more GDNF-stimulated axons to extend across the DREZ. It was also notable that the number of axons crossing the DREZ at 100 µm far exceeded the number of axons that entered the DREZ at 0 µm (i.e., astrocyte:PNS border) (*Figure 8H*). The number of intraspinal axons at 200 µm was also significantly greater than the number of axons crossing the DREZ at 100 µm. These findings suggest that elimination of myelin inhibitors enhanced intraspinal regeneration of

GDNF-stimulated axons, largely by promoting regenerative sprouting or additional branch formation of parental axons at and beyond the DREZ.

Collectively, these findings indicate that myelin inhibitors are indeed inhibitory to DR axons and can restrict their regeneration at the DREZ and beyond it within the spinal cord. They also demonstrate that genetic elimination of myelin inhibitors alone or combined with a conditioning lesion promotes minimal regeneration across the DREZ, reflecting their incomplete inhibitory effect. It can markedly enhance regeneration only in combination with a treatment that sufficiently enhances intrinsic growth capacity of DR axons.

### CSPG removal further enhances intraspinal regeneration of GDNF-stimulated axons in tKO mice

Lastly, we investigated the synergistic effect of CSPG degradation on intraspinal regeneration of GDNF-stimulated axons in *Rtn4/Mag/Omg* tKO mice. We microinjected LV-GDNF and LV-ChABC into dorsal horns and AAV2-GFP into DRGs at the time of L4 and L5 root crush (*Figure 9A*). At 2 wpi, wholemounts of GDNF/ChABC-expressed tKO mice revealed far broader areas of densely populated GFP+ axons growing into the CNS beyond the astrocyte:PNS border (*Figure 9C*, arrows) than in GDNF-expressed WT (*Figure 9B*, arrow) or in GDNF-expressed tKO mice (*Figure 8F*; see also *Figure 2—figure supplement 1*). Moreover, many GFP+ axons extended rostrocaudally close to the midline (*Figure 9C*, arrowheads), further evidence of enhanced intraspinal penetration and regeneration in GDNF/ChABC-expressed tKO mice. CS-56 antibody immunostaining confirmed effective removal of CSPGs in dorsal horns (*Figure 9E*). Serial transverse sections frequently showed GFP+ axons densely filling a broader and deeper area of the dorsal horn in GDNF/ChABC-expressed tKO mice (*Figure 9E, F*) than in GDNF-expressed WT (*Figure 9D*) or in GDNF-expressed tKO mice (*Figure 8G*; see also *Figure 2—figure supplement 1*). CGRP immunoreactivity on some sections was conspicuously bright, dense, and notably restricted to the superficial laminae, like CGRP immunoreactivity on the contralateral uninjured side (*Figure 9F*). Quantification indicated ~20% further increase in GFP+ axons crossing the DREZ at 100 µm, presumably reflecting enhanced regenerative sprouting, and ~60% increase in GFP+ axons at 400 µm in deeper portions of the dorsal horn in GDNF/ChABC-expressed tKO mice, compared to GDNF-expressed tKO mice (*Figure 9G*). These findings show that additional removal of CSPGs markedly and synergistically enhanced regeneration of GDNF-stimulated axons across the DREZ in *Rtn4/Mag/Omg* tKO mice. Therefore, like myelin inhibitors, CSPGs partially restrict DR axons at and beyond the DREZ, and their removal can significantly enhance regeneration across the DREZ only when combined with an intervention that sufficiently enhances intrinsic growth capacity of DR axons.

## Discussion

Using novel multifaceted strategies, we directly compared DR regeneration in WT, *Rtn4/Mag/Omg* tKO, ChABC-expressed tKO, GDNF-expressed tKO, and GDNF/ChABC-expressed tKO mice, either with or without a nerve conditioning lesion. Co-eliminating myelin inhibitors and CSPGs only slightly increased regeneration across the DREZ, surprisingly even after an additional conditioning lesion. Their absence, however, markedly and synergistically enhanced regeneration of GDNF-stimulated axons across and beyond the DREZ. These findings suggest the existence of a myelin/CSPG-independent mechanism that potently restrains most axons at the DREZ, and that this barrier can be overcome by elevating axon growth capacity, above that achieved by a nerve conditioning lesion. The results also suggest that therapeutically targeting both myelin inhibitors and CSPGs can enhance, but will not induce, regeneration of most DR axons into the spinal cord.

Various AAV vectors, including AAV2, demonstrate neuronal tropism (*Mason et al., 2010*; *Fischer et al., 2011*). Several AAV serotypes exhibit differential tropism for specific subpopulations of DRG neurons: AAV6 preferentially transduces small-sized neurons; AAV5, 8, and 9, large neurons (*Jacques et al., 2012*; *Yu et al., 2013*; *Wu et al., 2016*; *Kubota et al., 2019*). We found that AAV2 preferentially transduces NF+ neurons. Approximately 80% of GFP+ neurons were NF+, although only ~50% of lumbar DRG neurons are NF+ in mice (*Li et al., 2016*). IB4+ neurons constituted only ~5% of GFP+ neurons while CGRP+ neurons accounted for ~30%. Because ~30% of CGRP+ neurons also express NF200 (*Li et al., 2016*), our results are likely an overestimation of CGRP+ neurons projecting nociceptive axons. GFP expression in superficial dorsal laminae (I–IIi), where

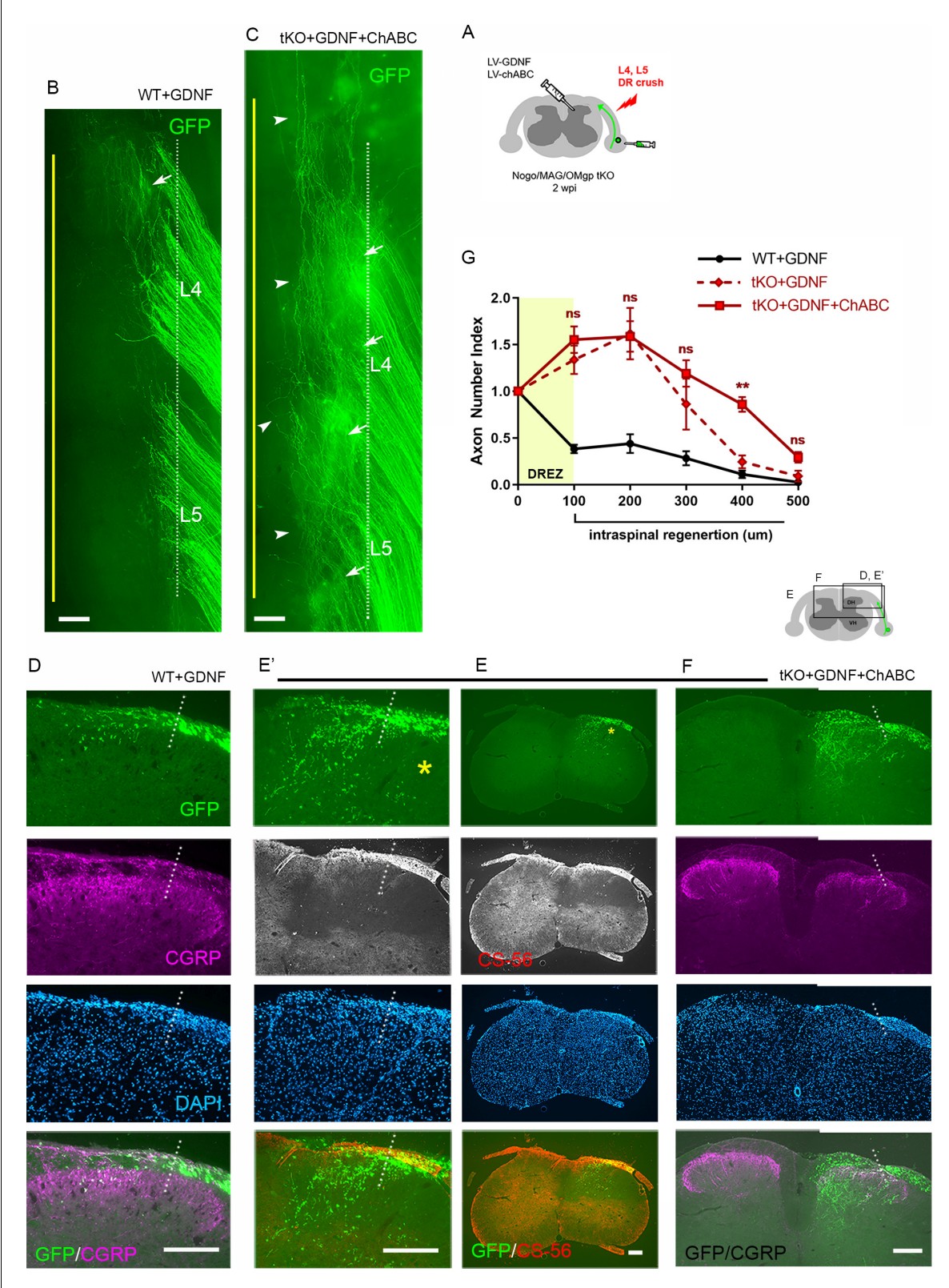

**Figure 9.** Chondroitin sulfate proteoglycan (CSPG) removal further enhances regeneration of glial cell line-derived neurotrophic factor (GDNF)-stimulated axons in triple knockout (tKO) mice. (**A**) Schematic illustration of the experimental procedures. LV-chondroitinase ABC (LV-ChABC) and LV-GDNF were injected into dorsal horn along the L4–L5 dorsal root entry zone (DREZ) in *Rtn4/Mag/Omg* tKO mice. (**B**) Wholemount view of a GDNF-expressed wildtype (WT) mouse. (**C**) Wholemount view of a ChABC/GDNF-expressed tKO showing broader areas of the DREZ and the CNS with

*Figure 9 continued on next page*

*Figure 9 continued*

densely accumulated GFP+ axons (arrows). Arrowheads denote numerous axons extending rostrocaudally close to the midline. (**D**) Transverse sections of a GDNF-expressed WT mouse. (**E–E'**) Transverse sections of a ChABC/GDNF-expressed tKO showing effective degradation of CSPGs confirmed by CS-56 immunoreactivity and many GFP+ axons densely filling broad and deep areas of the dorsal horn. (**F**) Transverse sections of a ChABC/GDNF-expressed tKO showing enhanced intraspinal regeneration of GFP+ axons and CGRP+ axons (magenta). CGRP+ immunoreactivity is bright, dense, and remarkably restricted to the superficial laminae. (**G**) Quantitative comparisons illustrating significantly more GFP+ axons in deeper portions of the dorsal horn in ChABC/GDNF-expressed tKO. 100 μm, p=0.5389, df = 210; 200 μm, p=0.9891, df = 210; 300 μm, p=0.2358, df = 210; 400 μm, p=0.0074, df = 210; 500 μm, p=0.5805, df = 210. Two-way repeated-measures ANOVA with Sidak's multiple comparisons test (ChABC/GDNF-expressed tKO: 19 sections, eight mice; GDNF-expressed tKO: 16 sections, five mice). ns: not significant. Scale bars = 200 μm (**B, C, D, E–E', F**).
The online version of this article includes the following source data for figure 9:

**Source data 1.** Source data for quantifying regeneration across the dorsal root entry zone.

---

nociceptive afferents terminate, was almost undetectable (*Figure 1E*, arrow). It appears, therefore, that the predominant transduction of NF+ neurons and minimal expression of GFP in CGRP+ axons make AAV2-GFP a valuable tracer of proprioceptive and mechanoreceptive axons.

Axon sparing has hampered consistent and conclusive studies with the widely used DR crush model (*Smith et al., 2012*). We found that complete crushing is difficult even in fluorescent mice whose axons are well visualized. Furthermore, improperly crushed axons often recover connectivity with distal axons (*Di Maio et al., 2011*; *Han et al., 2012*). Here, we studied L4–L5 roots because in our experience DR crush is more likely to spare cervical than lumbar roots. In addition, we used wholemount preparations to detect spared axons, which is possible because AAV2-GFP brightly labels numerous axons. Importantly, however, wholemount assessment does not guarantee complete lesions because viral transduction does not label all axons and spared axons that are deeply located may be overlooked.

Wholemounts also permitted simultaneous assessment of many axons in multiple roots, which has not been achieved previously. Greatly enhanced regeneration appeared as areas with unusually bright, dense GFP fluorescence. Rostrally extending bundles of GFP+ axons were also observed when intraspinal penetration exceeded 50%. These features require many intraspinally regenerating axons and therefore provide convincing evidence of robustly enhanced intraspinal penetration through the DREZ. Accordingly, wholemount assay is particularly useful for studies such as the present one, which uses a relatively small number of animals. We excluded mice for various reasons, including incomplete lesions and poor axon labeling, and used 3–8 mice per cohort to compare a total of 12 different animal groups. Nevertheless, no tKO, ChABC-expressed tKO, or conditioning lesioned/ChABC-expressed tKO mice exhibited either fluorescent areas of densely populated axons or rostral axon bundles. In contrast, these features were consistent findings in all the wholemounts of GDNF-expressed tKO and GDNF/ChABC-expressed tKO mice. Uniform absence or presence of these major features in wholemounts is therefore useful to determine if a specific factor serves as a major inhibitor or promoter of DR regeneration across the DREZ.

Our results suggest that myelin inhibitors and CSPGs play only a limited role in restraining axon outgrowth at the DREZ. For example, combined removal of these inhibitors enabled only about ~10% of axons to cross the DREZ and extend ~100 μm beyond it. Prolonged elimination for a month after injury also did not increase intraspinal penetration. More surprisingly, GDNF elicited markedly enhanced intraspinal regeneration in the absence of myelin inhibitors and CSPGs, unlike nerve crush conditioning lesion that produced little enhancement. If myelin inhibitors and CSPGs played a dominant role in restraining axons at the DREZ, then removing them should be sufficient to drive vigorous penetration. If their contribution is limited, then greater enhancement of axon growth potential might be essential for robust penetration, as we observed with supplementary expression of GDNF.

It is unlikely that the limited enhancement with a conditioning lesion was due to insufficient removal of Nogo/MAG/OMgp or CSPGs. Most notably, GDNF robustly enhanced regeneration in tKO and produced even more intraspinal regeneration in ChABC-expressed tKO mice. These observations, together with earlier studies that demonstrated efficacy of the *Rtn4/Mag/Omg* tKO and LV-ChABC (*Curinga et al., 2007*; *Lee et al., 2010*; *Jin et al., 2011*; *Han et al., 2017*), indicate that penetration through the DREZ was not limited because of insufficient elimination of Nogo/Reticulon-4, MAG, OMgp, CSPGs, and/or undigested CSPG carbohydrate stubs (*Lemons et al., 2003*). Our

findings are consistent with earlier observations in WT mice that degrading CSPGs alone (*Steinmetz et al., 2005*; *Wu et al., 2016*) or in combination with conditioning lesions (*Quaglia et al., 2008*) elicits minimal intraspinal penetration. Importantly, our findings demonstrate for the first time that the minimal penetration elicited by the absence of CSPGs was highly unlikely due to synergistic restraint by myelin inhibitors.

Our findings contrast with the robust, long-distance functional regeneration of myelinated DR axons reported in earlier studies that pharmacologically targeted Nogo receptor or CSPG/CSPG receptors, with or without a nerve crush conditioning lesion (*Harvey et al., 2009*; *Peng et al., 2010*; *Cheah et al., 2016*; *Yao et al., 2019*). Blocking inhibitors in adulthood may be more effective than eliminating them during development due to compensatory upregulation of other inhibitors (*El-Brolosy and Stainier, 2017*). It is noteworthy, however, that we removed CSPGs in adult tKO mice. Another possible explanation for the discrepancy is a species difference between mice and rats (*Lee and Lee, 2013*), but results have also conflicted in the same species. Additional myelin-associated inhibitors and ligands of Nogo and CSPG receptors have been identified (*Zhang et al., 2009b*; *Dickendesher et al., 2012*; *Thiede-Stan and Schwab, 2015*; *Ohtake et al., 2018*). However, CSPGs are the major additional ligands that bind to the Nogo receptor (*Dickendesher et al., 2012*), yet co-eliminating CSPGs and myelin inhibitors elicited little penetration through the DREZ. An alternative explanation is that these studies included incomplete lesions of varying extent, which were followed by acute or gradual functional recovery that did not require actual regeneration or relied on sprouting of spared axon terminals (*Cafferty et al., 2008*).

Myelin inhibitors and CSPGs are thought to stop axons at the DREZ by collapsing or trapping growth cones and inducing dystrophic endings (*Li et al., 1996*; *Golding et al., 1999*; *Ramer et al., 2001*; *Tom et al., 2004*; *Smith et al., 2012*). Our results revise this long-held view by suggesting the presence of inhibitory mechanism(s) of remarkably greater potency that act independently of myelin inhibitors and CSPGs. One such inhibitory mechanism may be the expression of additional repellent cues that are upregulated at the axotomized DREZ (*Andrews et al., 2009*; *Lindholm et al., 2017*). Another possibility is that most DR axons are stabilized at the DREZ by forming aberrant synapses rather than dystrophic endings with unknown postsynaptic cells (*Carlstedt, 1985*; *Liuzzi and Lasek, 1987*; *Stensaas et al., 1987*; *Di Maio et al., 2011*; *Son, 2015*). Consistent with this notion, DC axons form synapse-like endings on NG2+ cells following spinal cord injury (*Busch et al., 2010*; *Filous et al., 2014*). It remains to be determined whether NG2+ cells are postsynaptic cells at the DREZ, and whether all axons stop by forming aberrant synapses with NG2+ cells (*Son, 2015*). We could not examine this possibility in the present study because of the experimental challenges of differentiating dystrophic from synaptic endings and in blocking aberrant synapse formation. Other possible, although less likely, mechanisms include lack of a growth-permissive/promoting substrate (*Li et al., 2004*; *Han et al., 2017*; *Collins et al., 2019*) and/or inadequate intrinsic growth ability of DRG neurons (*Steinmetz et al., 2005*; *Nichols and Smith, 2020*).

Targeting Nogo, CSPGs, or their receptors in rodent models of spinal cord injury promotes minimal regeneration of myelinated axons, such as corticospinal tract (*Fink et al., 2015*; *Lang et al., 2015*; *Ito et al., 2018*). However, their removal can significantly enhance the growth promotion of a combinatory treatment by stimulating sprouting (*Wang et al., 2012*; *DePaul et al., 2017*; *Wu et al., 2017*). Consistent with these observations after spinal cord injury, we found that eliminating myelin inhibitors alone or in combination with CSPG removal markedly enhanced intraspinal regeneration of GDNF-stimulated axons, largely by stimulating regenerative sprouting. Thus, their removal can allow robust intraspinal regeneration but only when combined with a treatment that sufficiently enhances axon growth capacity. We chose to enhance growth capacity with GDNF rather than with zymogen or caBRAF, which stimulated greater intraspinal penetration (*Steinmetz et al., 2005*; *O'Donovan et al., 2014*). We were concerned that excessive enhancement of intrinsic growth capacity might impede evaluating the efficacy of extrinsic growth inhibitors in arresting axons at the DREZ. In addition, we observed that zymogen administration causes DRG neurons progressively to die in mice, likely due to severe inflammation (Kim et al., unpublished), and that it is challenging to provide supplemental genetic conditioning of caBRAF in tKO.

A single sciatic nerve conditioning lesion of DRG neurons has been extensively used for epigenetic, transcriptomic, and proteomic studies of peripheral and central regeneration (*Blesch et al., 2012*; *Li et al., 2015*; *Palmisano et al., 2019*). We found that a conditioning lesion does not sufficiently elevate axon growth capacity to overcome the potent myelin/CSPG-independent inhibitory

mechanism(s). Furthermore, a double conditioning lesion only modestly enhances intraspinal regeneration in WT mice, to a level comparable to that found with a single crush lesion in tKO mice, but that even the double lesion was far less efficacious than GDNF in enabling axons to cross the DREZ. Therefore, more axons may be able to cross the DREZ when the double conditioning lesion is combined with co-eliminating myelin inhibitors and CSPGs, but we will expect to observe substantially fewer axons than when GDNF treatment is combined with co-eliminating these inhibitors. Although our results highlight the importance of sufficiently elevating axon growth capacity, GDNF enhanced penetration of ~40% DR axons in WT mice. Moreover, the intraspinal regeneration it elicited in the absence of myelin inhibitors and CSPGs was apparently enhanced but did not appear sufficiently robust to power the long-distance regeneration required for proprio-/mechanoreceptive axons. The DR regeneration induced by several other neurotrophic factors, cytokines, and downstream effectors has been generally weaker than that elicited by GDNF in ChABC-expressed tKO mice (*O'Donovan et al., 2014*; *Liu et al., 2016*; *Wu et al., 2016*). Learning therapeutically applicable treatment of maximizing axon growth capacity must be as important as understanding the myelin/CSPG-independent mechanism that powerfully halts axons at the DREZ, and perhaps also within the damaged spinal cord.

# Materials and methods

## Key resources table

| Reagent type (species) or resource | Designation | Source or reference | Identifiers | Additional information |
|---|---|---|---|---|
| Strain, strain background (*Mus musculus*) | *Rtn4/Omg* double knockout | Dr. Binhai Zheng (UCSD) *Lee et al., 2010* | RRID:MGI:3624445 RRID:MGI:3821705 | |
| Strain, strain background (*Mus musculus*) | *Mag* knockout | Dr. Jae K. Lee (U.Miami) | Stock # 006865; RRID:IMSR JAX:006865 | |
| Strain, strain background (*Mus musculus*) | C57BL/6J | The Jackson Laboratory | Stock # 000664; RRID:IMSR JAX:000664 | |
| Other | Vectashield | Vector Laboratories, Burlingame | H-100 RRID:AB2336789 | Mounting medium |
| Transfected construct (*Mus musculus*) | scAAV2-eGFP | Dr. George M. Smith (Temple University) | | AAV construct to transfect and express eGFP (*Liu et al., 2014*) |
| Transfected construct (*Mus musculus*) | LV-chABC | Dr. George M. Smith (Temple University) | | Lentiviral construct to transfect and express chABC (*Curinga et al., 2007*) |
| Transfected construct (*Mus musculus*) | LV-GDNF | Dr. George M. Smith (Temple University) | | Lentiviral construct to transfect and express GDNF (*Zhang et al., 2013*) |
| Antibody | Anti-NF200 (rabbit polyclonal) | Sigma | #N4142 RRID:AB477272 | IHC (1:500) |
| Chemical compound, drug | Isolectin B4, biotin conjugate | Sigma | #L2140 RRID:AB2313663 | IHC (1:200) |
| Antibody | Anti-CGRP (rabbit polyclonal) | Peninsula Labs | #T4032 RRID:AB2307330 | IHC (1:2000) |
| Antibody | Anti-GFP (mouse monoclonal) | Avés Labs Inc | #GFP-1020 RRID:AB10000240 | IHC (1:500) |
| Antibody | Anti-GFAP (rabbit polyclonal) | Agilent | #N1506 RRID:AB10013482 | IHC (1:500) |
| Antibody | Alexa Fluor 647-goat anti-rabbit IgG secondary antibody | Invitrogen | #31573 RRID:AB2536183 | IHC (1:400) |

*Continued on next page*

*Continued*

| Reagent type (species) or resource | Designation | Source or reference | Identifiers | Additional information |
|---|---|---|---|---|
| Antibody | Fluorescein (FITC)-conjugated goat anti-rabbit IgG secondary antibody | Millipore | #AP307F RRID:AB92652 | IHC (1:400) |
| Antibody | Alexa Fluor 568-conjugated goat anti-mouse IgG$_1$ secondary antibody | Invitrogen | #A21124 RRID:AB141611 | IHC (1:400) |
| Antibody | Rhodamine (TRITC)-conjugated streptavidin secondary antibody | Jackson ImmunoResearch Labs Inc | #016-020-084 RRID:AB2337237 | IHC (1:400) |
| Antibody | Alexa-Fluor 568-conjugated goat anti-rabbit secondary antibody | Invitrogen | #A11011 RRID:AB143157 | IHC (1:400) |
| Antibody | Alexa Fluor 488-donkey anti-chicken IgG secondary antibody | Jackson ImmunoResearch Labs Inc | #703-545-155 RRID:AB2340375 | IHC (1:400) |
| Other | DAPI stain | Thermo Fisher Scientific | #D1306 RRID:AB2629482 | IHC (1:1000) |
| Other | Nissl substance stain | Thermo Fisher Scientific | #N21482 RRID:AB2620170 | IHC (1:200) |
| Software, algorithm | MetaMorph Image Analysis Software | Molecular Devices | RRID:SCR002368 | |
| Software, algorithm | AxioVision Imaging System | Zeiss | RRID:SCR002677 | |
| Software, algorithm | Axio Imager | Zeiss | RRID:SCR018876 | |
| Software, algorithm | Imaris | Bitplane | RRID:SCR007370 | |
| Software, algorithm | Adobe Photoshop | Adobe Inc | RRID:SCR014199 | |
| Software, algorithm | PRISM 8.0 | GraphPad | RRID:SCR002798 | |

## Mice

All animal care and procedures were conducted in accordance with the National Research Council's Guide for the Care and Use of Laboratory Animals and approved by the Institutional Animal Care and Use Committee at Lewis Katz School of Medicine at Temple University (animal protocol #4919), Philadelphia, PA, USA. Congenic *Rtn4/Mag/Omg* tKO mice were generated by breeding congenic lines of *Rtn4/Omg* double knockout mice obtained from Dr. Binhai Zheng (University of California at San Diego) and of MAG single knockout mice obtained from Dr. Jae K. Lee (University of Miami). Male and female, 2–3-month-old *Rtn4$^{-/-}$/Omg$^{-/-}$/Mag$^{-/-}$* mice and age-matched C57BL/6J mice (The Jackson Laboratory) were used. Mice were genotyped at the time of weaning by tail clipping and PCR analysis using the primers described in earlier studies (*Li et al., 1994*; *Lee et al., 2010*).

## Dorsal root crush

Mice were anesthetized with an intraperitoneal injection of xylazine (8 mg/kg) and ketamine (120 mg/kg). Supplements were given during the procedure as needed. Following a 2- to 3-cm-long incision in the skin of the back, the spinal musculature was reflected and the L3–S1 spinal cord segments exposed by hemilaminectomies to prepare for unilateral L4–L5 root crush. The cavity made by the laminectomies was perfused with warm sterile Ringer's solution. A small incision was made in the dura overlying the L4 and L5 DR. A fine forceps (Dumont #5; Fine Science Tools, Foster City, CA) was introduced subdurally and the DR crushed for 10 s. To avoid scar formation and possible compression, we applied a piece of thin SILASTIC membrane (Biobrane, Bertek Pharmaceuticals, Sugarland, TX) over the laminectomy site and covered it with a layer of thicker artificial dura (Gore Preclude MVP Dura Substitute, W.L. Gore and Associates, Flagstaff, AZ). The overlying musculature was closed with 5-0 sutures, and the skin closed with wound clips. All animals received subcutaneous injections of saline (0.9% w/v NaCl) and buprenorphine (0.05 mg/kg) for postoperative pain

management and remained on a heating pad until fully recovered from anesthesia. Mice were perfused 2 or 4 weeks after the crush, and axon regeneration was analyzed in cryostat sections or in thin dorsal slice preparations of whole spinal cord.

## Nerve conditioning lesion

DRG neurons were preconditioned by crushing the sciatic nerve in the lateral thigh of the ipsilateral hind leg 10 days before the DRs were crushed. Animals were anesthetized as described above; the skin and superficial muscle layer of the midthigh were opened; and the sciatic nerve was crushed for 10 s with fine forceps (Dumont #5; Fine Science Tools). The muscle and skin were then closed in layers and the animals allowed to recover on a heating pad until fully awake. For a double conditioning lesion, the sciatic nerve was completely transected with microscissors (Fine Science Tool) 3 days before the unilateral L4–L5 root crush and again, ~1 cm proximal to the site of the first transection, 7 days after DR crush. This sequence of a double conditioning lesion paradigm has been reported to enhance regeneration of DRG axons ascending in the DCs after spinal cord injury (*Neumann et al., 2005*).

## Production and microinjection of AAV2-GFP, LV-chABC, and LV-GDNF

Recombinant self-complementary adeno-associated virus 2 (scAAV2) carrying eGFP was generated by helper virus-free system (*Ayuso et al., 2010*) as described previously (*Liu et al., 2014*). Replication-deficient lentiviruses encoding either chABC (LV-chABC) or GDNF (LV-GDNF) used a pBOB lentiviral expression vector with CMV-enhanced chicken β-actin (CAG) promoter. The procedures to generate the viruses and their efficacy were described before: LV-chABC (*Curinga et al., 2007*; *Jin et al., 2011*; *Han et al., 2017*); LV-GDNF (*Zhang et al., 2009a*; *Deng et al., 2013*; *Zhang et al., 2013*; *Kelamangalath et al., 2015*; *Han et al., 2017*). AAV2-GFP was microinjected into lumbar DRGs using a micropipette pulled to a diameter of 0.05 mm and a nanoinjector (World Precision Instruments, Sarasota, FL). The viruses were injected at the time of DR crush. For each injection, the micropipette was introduced 0.5 mm into the DRG and a total volume of AAV (0.8 µl) containing $>2 \times 10^{12}$ GC/ml injected over a 10 min period. The glass needle was left in place for 2 min after each injection. For LV-GDNF, five injections of 0.3 µl lentivirus (a total of $2 \times 10^7$ viral particles) were equally spaced rostrocaudally along the L4–L5 DREZ. Injections were made at a rate of 100 nl/min at a depth of 0.25 mm from the spinal cord dorsal surface.

## Tissue processing, selection, and immunohistochemistry

Two or four weeks after DR crush, mice were sacrificed by an overdose of Euthasol and perfused transcardially with 0.9% saline followed by 4% paraformaldehyde (PFA) in 0.1 M phosphate buffer (PBS, pH=7.4). The spinal cords with attached DRs and DRGs were removed and examined in wholemounts to exclude those tissues with poor transduction of AAV2-GFP and/or spared axons. Properly lesioned and GFP+ tissues were first examined in wholemounts to assess DR regeneration across the DREZ. Tissues were then processed for analysis on cryostat sections. One or two representative tissues were processed for immunohistochemistry in spinal cord wholemounts. For analysis on cryostat sections, spinal cords were post-fixed in 4% PFA overnight at 4°C and cryoprotected in 30% sucrose for 1–2 days. The tissues were then embedded in M-1 Embedding Matrix (Thermo Fisher Scientific, Waltham, MA), transversely sectioned at 20 µm using a cryostat (Leica Microsystems, Germany), and mounted directly on slides (Superfrost Plus, Fisher Scientific, Pittsburgh, PA). For immunolabeling, sections were rinsed three times in PBS for 30 min followed by 10 min incubation with 0.1 M glycine in PBS, and 15 min incubation with 0.2% Triton X-100, 2% bovine serum albumin (BSA) in PBS (TBP). Sections were incubated with primary antibodies overnight at 4°C, washed three times for 30 min with 2% BSA in PBS, and incubated with secondary antibodies for 1 hr at room temperature. After three rinses with PBS, sections were mounted in Vectashield (Vector Laboratories, Burlingame, CA) and stored at –20°C until examination. For wholemount immunostaining of spinal cords, spinal cords with attached DRs were post-fixed in 4% PFA for 2 hr at 4°C and the dura mater removed. The spinal cord was then rinsed three times with PBS for 30 min, incubated for 10 min in 0.1 M glycine in 2% BSA/PBS, and permeabilized with pre-cooled methanol at –20°C for 10 min. After extensive rinsing in PBS, the spinal cord was incubated with primary antibody diluted in TBP at room temperature overnight, rinsed thoroughly in TBP the following day, and incubated with secondary antibodies for

2 hr at room temperature. After rinsing in PBS, a thin slice of dorsal spinal cord (~2 mm thick) with attached DR stumps was cut with a microscissor and mounted on slides in Vectashield (Vector Laboratories).

## Antibodies

Primary antibodies were used at the following concentrations for immunohistochemistry: rabbit anti-NF200 (1:500, Sigma, St. Louis, MO, #N4142), IB4-biotin conjugate (1:200, Sigma, #L2140), rabbit anti-CGRP (1:2000, Peninsula Labs, San Carlos, CA, #T4032), chicken anti-GFP (1:1000, Avés Labs Inc, Davis, CA, #GFP-1020), and rabbit anti-GFAP (1:500, Agilent, Santa Clara, CA, #N1506). Secondary antibodies used were Alexa Fluor 647-goat anti-rabbit IgG (1:400, Invitrogen, Indianapolis, IN, #31573), fluorescein (FITC)-conjugated goat anti-rabbit IgG (1:400, Millipore, Temecula, CA, AP307F), Alexa Fluor 568-conjugated goat anti-mouse IgG1 (1:400, Invitrogen, A21124), Alexa-Fluor 568-conjugated goat anti-rabbit (1:400, Invitrogen, #A11011), rhodamine (TRITC) streptavidin (1:400, Jackson ImmunoResearch Labs Inc, West Grove, PA, #016-020-084), or Alexa Fluor 488-donkey anti-chicken IgG (1:400, Jackson ImmunoResearch Labs Inc, #703-545-155). DAPI nucleic acid stain (1:1000, Thermo Fisher Scientific, Pittsburgh, PA, #D1306) for cell nuclei or Nissl substance stain (1:200, Thermo Fisher Scientific, #N21482) for neuronal cells was used to counterstain prior to final wash steps in PBS.

## Microscopy and image acquisition

An Olympus BX53 microscope equipped with Orca-R2 CCD camera (Hamamatsu, Japan) controlled by MetaMorph Image Analysis Software (Molecular Devices, San Jose, CA) was used to examine serial cryostat sections and dorsal wholemounts. Z stacked images were acquired using the Axio Imager (Zeiss, Germany) upright fluorescence microscope and AxioVision Imaging System (Zeiss, Germany) software or a SP8 confocal microscope (Leica Microsystems, Buffalo Grove, IL). All images were processed using Imaris (Bitplane, Windsor, CT) and Adobe Photoshop (Adobe Inc, San Jose, CA).

## Analysis of regeneration across the DREZ

The location of the DREZ, where DR axons stop their regeneration at the entrance of spinal cord, was demarcated by GFAP immunostaining of astrocytes. The white dotted lines (e.g., *Figure 1D, E*) denote the location of the outer (peripheral) boundary of the DREZ, as defined by GFAP immunolabeling of astrocytic processes that extend peripherally (termed here 'astrocyte:PNS border' or 'astrocytic border'). When GFAP was not immunolabeled, the astrocyte:PNS border was identified based on the greater abundance of cell nuclei in the PNS than CNS. For comparative evaluation of regeneration across the DREZ, we considered that DR axons penetrated into the CNS when they extended at least 100 µm beyond the astrocyte:PNS border. For quantitative analysis, digital images were captured from 3 to 6 representative, non-adjacent sections taken from L4 and L5 segments of each mouse (n = 3–8 mice per cohort). A raw image was converted to a binary image using ImageJ with a threshold that appropriately separated GFP and background fluorescence. Lines were drawn at 100 µm before the border (in PNS territory), at the DREZ outer border, and at 100 µm intervals into CNS territory. The number of intersections of GFP+ axons at these distances was counted. The number of axons that crossed the outer border was normalized by the number of GFP+ axons counted at 100 µm before the border in the PNS, and then averaged by the number of evaluated sections and animals. This quantification resulted in the 'axon number index' that indicated the relative number and distance of axons that regenerated into the CNS in each group of mice.

## Statistical analysis

All statistical analyses were performed using PRISM 8.0 (GraphPad, San Diego, CA). Statistical analysis was by two-way repeated-measures analysis of variance (ANOVA) with Tukey's multiple comparison tests. All data are presented as mean ± SD. Results were considered statistically significant if the p-value was < 0.05.

## Acknowledgements

We thank Drs. Alan Tessler, Matt Grove, and Harun Noristiani for critical reading of the manuscript. We greatly thank Dr. Binhai Zheng for *Rtn4/Mag/Omg* triple and *Rtn4/Omg* double knockout mice, Dr. Jae Lee for MAG mutant mice, and Yingpeng Liu for technical assistance with LV-chABC and LV-GDNF. This work was supported by NIH NINDS (NS079631) and Shriners Hospitals for Children (86600, 84050) to Y-JS.

## Additional information

### Funding

| Funder | Grant reference number | Author |
| --- | --- | --- |
| National Institute of Neurological Disorders and Stroke | NS079631 | Young-Jin Son |
| Shriners Hospitals for Children | 86600 | Young-Jin Son |
| Shriners Hospitals for Children | 84050 | Young-Jin Son |

The funders had no role in study design, data collection and interpretation, or the decision to submit the work for publication.

### Author contributions

Jinbin Zhai, Data curation, Formal analysis, Writing - original draft; Hyukmin Kim, Formal analysis, Investigation, Methodology, Writing - original draft; Seung Baek Han, Investigation, Methodology; Meredith Manire, Formal analysis, Investigation; Rachel Yoo, Validation, Investigation; Shuhuan Pang, Resources, Investigation; George M Smith, Resources, Methodology; Young-Jin Son, Conceptualization, Supervision, Funding acquisition, Investigation, Writing - original draft, Project administration, Writing - review and editing

### Author ORCIDs

Young-Jin Son https://orcid.org/0000-0001-5725-9775

### Ethics

Animal experimentation: All animal care and procedures were conducted in accordance with the National Research Council's Guide for the Care and Use of Laboratory Animals and approved by the Institutional Animal Care and Use Committee at Lewis Katz School of Medicine at Temple University, Philadelphia, PA, USA. (animal protocol #4919).

### Decision letter and Author response

Decision letter https://doi.org/10.7554/eLife.63050.sa1
Author response https://doi.org/10.7554/eLife.63050.sa2

## Additional files

### Data availability

Numerical data generated or analyzed during this study are included in the manuscript and supporting files. Source data files have been submitted for Figures 3, 4, 5,6, 6-S1, 7, 8, 9.

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
