## [Decision Letter]

**Acceptance summary:**

The reasons why axon regeneration into the spinal cord after a dorsal root crush are still not fully understood. In this manuscript, Zhai et al., report that co-eliminating myelin inhibitors and chondroitin sulfate proteoglycans elicited regeneration of only a few conditioning-lesioned dorsal root axons across the dorsal root entry zone, but markedly and synergistically enhanced regeneration of GDNF-stimulated axons. This work provides evidence that myelin inhibitors and chondroitin sulfate proteoglycans are not the primary mechanism stopping axons at the dorsal root entry zone.

**Decision letter after peer review:**

[Editors’ note: the authors submitted for reconsideration following the decision after peer review. What follows is the decision letter after the first round of review.]

Thank you for submitting your work entitled "Co-targeting myelin inhibitors and CSPGs enhances sensory axon regeneration within, but not into, the spinal cord" for consideration by *eLife*. Your article has been reviewed by 3 peer reviewers, and the evaluation has been overseen by a Reviewing Editor and a Senior Editor. The following individuals involved in review of your submission have agreed to reveal their identity: Veronica J Tom (Reviewer #2); Jerry Silver (Reviewer #3).

Our decision has been reached after consultation between the reviewers. Based on these discussions and the individual reviews below, we regret to inform you that your work will not be considered further for publication in *eLife*.

There was consensus that the manuscript addressed an important question of the role of myelin inhibitors and CSPGs in inhibiting sensory axon regeneration through the DREZ after injury. Moreover, conflicting findings have been reported and have used varying methods to injure and track regeneration. However, reviewers thought that whereas the confirmation that myelin inhibitors and CSPGs are not involved in sensory axon regeneration through the DREZ is important, it does not represent a major advance in the field. In addition, several concerns have been also raised about the current results, including using GFAP as the only marker for the boundary between the CNS and PNS. There was also a concern that the current data do not support the claim that CSPG is insufficient to enable axons to cross the DREZ. Given that addressing these and other concerns described in the individual reviews below, large amount experimental work would be required, your manuscript will not be considered further.

*Reviewer #1:*

This is an interesting article by Zhai et al., aimed at elucidating the role of myelin inhibitors and CSPGs in inhibiting sensory axon regeneration through the DREZ after injury. There are several studies that have investigated this in the past, but they have conflicting findings and have used varying methods to injure and track regeneration. In this manuscript, the authors want to determine the respective roles of myelin inhibitors and CSPGs in this process, and whether they also control regrowth of axons once they migrate through the DREZ. The question is interesting and important given that this form of peripheral regeneration is not successful, which is in contrast to the regeneration we see in other places in the PNS. However, while the authors have used a novel method of tracking axonal regrowth in this manuscript, their data is by in large confirmatory of what is already known and does not represent an advance (the author cites all of this themselves). The data that represents some advancement is in the last 2 figures. However, it is very preliminary and in my opinion, is not yet ready for publication. Below, I will also list a few concerns.

1. While the confirmation that myelin inhibitors and CSPGs are not involved in sensory axon regeneration through the DREZ is important, it does not represent a major advance in the field.

2. Also, I have major concerns with most of the figures in that many do not show data for both WT and experimental (tKO, etc.). The quantified data is present, but in my opinion, images need to also be shown, especially when one of the two conditions is included. This allows the reader to assess the images themselves before accepting that the quantified data matches.

3. Another major concern is the use of GFAP as the only marker for the boundary between the CNS and PNS. There are other makers, laminin is an example, that would give a much clearer delineation of this boundary. And in some studies/data, they don't use anything. This brings into question rigor, and I think it's essential to show this boundary given that the premise of the manuscript is to determine if axons can cross it.

4. The new data in this field is only represented in the final 2 figures, and it is preliminary. Additional studies describing the possible axon branching, etc., would make these data stronger.

*Reviewer #2:*

The reasons why axon regeneration into the spinal cord after a dorsal root crush are still not fully understood. In this manuscript, Zhai et al., set out to determine the role of myelin inhibitors and chondroitin sulfate proteoglycans (CSPG) in limiting axon regeneration across the dorsal root entry zone (DREZ) after a complete dorsal root crush injury. To do this, they performed L4 and L5 dorsal root crush injuries in transgenic mice that lacked the myelin proteins Nogo, MAG, and OMgp (triple knockout). In some of these animals, they injected a lentivirus for chondroitinase ABC to digest CSPGs. They also performed conditioning lesions and expression of GDNF with lentivirus. To identify axons, mainly proprioceptive and mechanoreceptive, they injected AAV2-GFP into the L4 and L5 dorsal root ganglia.

Interestingly, they found that the triple knockout did not result in enhanced axon regeneration into the spinal cord. Expression of ChABC or the conditioning lesion modestly improved the axon regeneration in the triple knockout animals. There did not appear to be a synergistic effect, as the combination of the conditioning lesion and ChABC did not increase axon regeneration further. Expression of GDNF did increase growth into the spinal cord in both WT and the triple knockout animals.

The manuscript is well written, and the data are clearly presented. There are a few comments to strengthen the authors' conclusions:

1. This group previously showed that, after a dorsal root crush, the axons form synaptic contacts on NG2 cells that was thought to limit axon regrowth. Was the degree of this impacted in any of the groups in which they observed enhanced growth into the spinal cord? The authors allude to a synapse-based inhibitory mechanism in the discussion and refer to a manuscript that is in preparation. However, examining the number of synapses formed by these axons in the difference conditions would strengthen the data here.

2. The GDNF data are interesting. The authors conclude that GDNF enabled "intraspinal penetration and regeneration". However, as presented, it is not clear that the increase in axons within the cord is due to more neurons extending an axon across the DREZ with GDNF or sprouting of axons from the same number of neurons that had already crossed the DREZ? Either would lead to more axons within the cord but by different mechanisms.

3. The authors conclude that the absence of CSPG is insufficient to enable axons to cross the DREZ. Their data do not support this claim, as they did see more axons within several hundred microns in the spinal cord when ChABC was expressed (Figure 4). While it is apparent that CSPGs are not the only mechanism limiting growth across the DREZ, as discussed by the authors, their data do support that CPSGs play a role in limiting growth into the spinal cord. The conclusions should be modified accordingly.

*Reviewer #3:*

The authors have described a rigorous set of experiments where they systematically removed singly or in combination, purported inhibitors of axon regeneration as well as the further addition of a single crush conditioning lesion (to promote the intrinsic capacity for growth) in order to attempt to promote regeneration of adult sensory axons across the DREZ. In the end, they found that only GDNF treatment (as has already been documented) of the DREZ (but not removal of myelin or CSPG inhibitors nor crush conditioning) was sufficient to get axons across the barrier. However, it was shown convincingly that additional removal of inhibitors allowed for more lengthy regeneration within the CNS compartment. In general, this is a well-documented series of experiments that sheds new light on the mechanisms that are at work in curtailing sensory axon regeneration at one of the most inhibitory regeneration boundaries within the nervous system, the DREZ. I have made a variety of suggestions for revisions of the text in order to soften, at least a bit, the sometimes overly zealous statements about CSPGs and myelin inhibitors. There is also a suggestion about critically altering the discussion as well as the possibility of adding a series of additional experiments (although, following the *eLife* guidelines, the additional experiments are deemed not to be absolutely required). My comments and suggestions for rewriting are listed in order below.

Pg 3. This potent blockade is surprising…the growth potential of a relatively small number of DRG neurons to penetrate a glial scar…

Pg 3. CSPG removal, however, when combined with chronic, inflammation induced conditioning lesions or neurotrophic factors, has significantly enhanced intraspinal regeneration of DR axons (Steinmetz et al., 2005; Wu et al., 68 2016; Guo et al., 2019).

Pg. 3 These reports have supported a contributing inhibitory role of CSPGs by suggesting that removing CSPGs alone only modestly increases regeneration presumably because myelin inhibitors are sufficient to stop axons at the DREZ.

Pg. 3 Despite the long-held view by some attributing…. to myelin inhibitors by themselves.

Pg.3 …how robustly unconditioned or lesion induced conditioned DR axons regenerate….

Pg. 4 …CSPGs and addition of a trauma induced conditioning lesion,….

Pg 4 Thus, myelin inhibitors and CSPGs individually are not crucial factors….

Pg 8 …that by themselves might be capable of arresting axons at the DREZ.

Pg 8 … due to redundant and potent inhibition by other inhibitors

Further discussion about enhanced axons at the DREZ when lenti ch'ase was delivered is warranted. Papers of David Muir are relevant here.

Pg 10 Instead, these findings suggest that neither myelin inhibitors nor CSPGs that reside in the vicinity of the DREZ after injury are, by themselves, sufficiently potent to prevent regeneration of DR axons at the DREZ.

Pg 10 DR axons fail to penetrate the DREZ in Nogo/MAG/OMgp tKO even after a nerve crush conditioning lesion

Pg 11 Therefore, removal of myelin associated inhibitors does not enable even crush conditioned axons to regenerate beyond the DREZ.

Somewhere In the discussion I think it is critical for the authors to compare their single lesion induced conditioning results to those of Steinmetz, et al., 2005 who used week-long zymosan conditioning to induce a maximally robust conditioning lesion response. The authors should note that in Steinmetz et al., in vitro, a single surgical crush conditioning lesion (like that used in the present experiments) did not allow for regeneration of DRG axons across an inhibitory CSPG gradient (see Figure 1 in Steinmetz et al., 2005). It would have been interesting if the authors had used a double conditioning lesion paradigm (as in Neumann et al., PNAS 2005) to see if maximal surgical conditioning could enhance regeneration after elimination of one or both of the other purported axon growth inhibitors. If the authors are willing to perform the double conditioning experiments then this could enhance the paper but it is not being suggesting that this is essential. However, the lack of double conditioning experiments does somewhat dampen enthusiasm for the paper. It is being suggested that, at a minimum, the authors surely should discuss the possibility that their single conditioning paradigm is not optimal.

Pg 17 When the authors discuss the aberrant synapse hypothesis to explain regeneration failure at the DREZ they should also, of course, cite and discuss the published work by Busch et al., J Neurosci 2010 and Filous et al., J Neurosci 2014, who described this phenomenon in the CNS.

---

## [Author Response]

[Editors’ note: The authors appealed the original decision. What follows is the authors’ response to the first round of review.]

There was consensus that the manuscript addressed an important question of the role of myelin inhibitors and CSPGs in inhibiting sensory axon regeneration through the DREZ after injury. Moreover, conflicting findings have been reported and have used varying methods to injure and track regeneration. However, reviewers thought that whereas the confirmation that myelin inhibitors and CSPGs are not involved in sensory axon regeneration through the DREZ is important, it does not represent a major advance in the field.

As explained in greater detail in the response to reviewer #1 (see below), the current literature indicates that both myelin inhibitors and CSPGs play a “major” role in inhibiting regenerating axons at the DREZ, rather than that they are “not involved”. Our finding that simultaneous removal of both myelin inhibitors and CSPGs does NOT elicit robust regeneration of axons across the DREZ contradicts the widely accepted notion in the field, rather than “confirms” it, and therefore “sheds new light into the mechanisms,” as reviewer #3 correctly commented.

It is also noteworthy that the current assumption is based on earlier studies that ‘individually’ targeted myelin inhibitors or CSPGs, rather than “simultaneously” eliminating both of them as we did in the present work. Simply stated, no similar combinatorial study has been published before. Our work is the first to demonstrate the direct and surprisingly minimal consequences of eliminating both types of extrinsic inhibitors on regeneration across the DREZ. Moreover, we also boosted intrinsic axon growth capacity with a nerve conditioning lesion while eliminating myelin inhibitors and CSPGs. This triple therapy, combining myelin inhibitors, CSPGs and conditioning lesion, is also novel, and provides additional new and important insights into mechanisms and therapy.

Some of these individual targeting studies reported “robust” anatomical and functional regeneration across the DREZ, which, in our opinion, was overstated or unconvincing. In the present work, we introduced a novel wholemount assay and selective viral tracing of proprio- and mechanoreceptive axons to exclude inappropriately operated animals and to evaluate more rigorously the robustness of DR regeneration across the DREZ. Because incomplete lesions with axon sparing have been a major concern in the fields of both CNS and PNS regeneration (Steward et al., 2012), and because our injury model (dorsal root crush) is widely used for studying spinal cord regeneration, our paper also has an additional, and broadly significant, technical impact.

Our work also shows, for the first time, that co-eliminating myelin inhibitors and CSPGs markedly and synergistically enhances GDNF-evoked intraspinal regrowth of DR axons. These findings are interesting, as all the reviewers indicated, and have important therapeutic implications: interventions targeting myelin inhibitors and CSPGs can enhance, but they will not induce, “robust” intraspinal regeneration of primary sensory axons. We consider, therefore, that the conceptual and technical advance of our work will be more important than just disputing the entrenched, conventional view in the field, and that the study will move the field in a new and important direction.

In addition, several concerns have been also raised about the current results, including using GFAP as the only marker for the boundary between the CNS and PNS. There was also a concern that the current data do not support the claim that CSPG is insufficient to enable axons to cross the DREZ. Given that addressing these and other concerns described in the individual reviews below, large amount experimental work would be required, your manuscript will not be considered further.

We thank the reviewers for raising these valuable and constructive comments. As detailed below, we believe that we have been able to address all the concerns of the reviewers.

Reviewer #1:This is an interesting article by Zhai et al., aimed at elucidating the role of myelin inhibitors and CSPGs in inhibiting sensory axon regeneration through the DREZ after injury. There are several studies that have investigated this in the past, but they have conflicting findings and have used varying methods to injure and track regeneration. In this manuscript, the authors want to determine the respective roles of myelin inhibitors and CSPGs in this process, and whether they also control regrowth of axons once they migrate through the DREZ. The question is interesting and important given that this form of peripheral regeneration is not successful, which is in contrast to the regeneration we see in other places in the PNS. However, while the authors have used a novel method of tracking axonal regrowth in this manuscript, their data is by in large confirmatory of what is already known and does not represent an advance (the author cites all of this themselves). The data that represents some advancement is in the last 2 figures. However, it is very preliminary and in my opinion, is not yet ready for publication. Below, I will also list a few concerns.1. While the confirmation that myelin inhibitors and CSPGs are not involved in sensory axon regeneration through the DREZ is important, it does not represent a major advance in the field.

We thank the reviewer for this comment and for finding our work interesting and methodologically novel. We regret a number of statements that seem to have caused this reviewer to underestimate the novelty and significance of our data. As suggested, we have clarified those statements and explained our work more directly.

The reviewer’s comment that “…confirmatory…not represent an advance (the author cites all of this themselves)” seems in part due to our statements of “consistent with earlier studies or previous observations”. These statements are primarily focused toward methodological similarities not experimental results observed. We would like to emphasize that none of these earlier studies reported the same or similar findings as the main results that we present in each figure (see below).

In the Results section, we referred to: (1) earlier studies of “cross sections” to validate our novel “wholemount” assay, (2) earlier studies of lentivirus-ChABC to validate our successful uses of the same lentivirus, (3) earlier studies of “WT mice” to validate our first-time evaluation of Nogo/MAG/OMgp “tKO mice”, and (4) an earlier study of GDNF-expressed “WT mice” to validate our uses of lentivirus-GDNF and the results we obtained by their first-time application in “tKO mice”.

In the Discussion section, we referred to earlier reports of minimal rather than robust regeneration following CSPG removal and/or conditioning lesions in “WT mice”, consistent with our results in “tKO mice”, in which we observed minimal regeneration even after additional removal of myelin inhibitors. The consistency of these results, however, is not a principal finding of the experiment. The novel and much more important aspect of the result is that myelin inhibitors are not responsible for keeping the regeneration minimal, although some in the field have speculated that myelin inhibitors compensate for the absence of CSPGs, which we do not observe.

Our data actually contradict, not confirm, the prevailing view in the field. “What is already known” is that both myelin inhibitors and CSPGs are major factors preventing regeneration across the DREZ, again which we do not observe.

Importantly, however, this notion derives from studies that “individually” targeted myelin inhibitors or CSPGs. Two groups (Eric Frank’s and Marina Mata’s) reported that pharmacological targeting of myelin signaling “dramatically” enhanced DR regeneration past the DREZ. No published studies have contradicted these findings, and these reports continue to be cited in support of the crucial or “major” role played by myelin inhibitors at the DREZ.

Different investigators have reported equally dramatic anatomical and functional regeneration due to pharmacological and individual targeting of CSPGs or a nerve conditioning lesion (e.g., Chong et al., 1999; Yao et al., 2019). In contrast to the case for myelin inhibitors, however, other groups, including ours (George Smith and YJ Son), have studied CSPGs and nerve conditioning lesions, and reported that eliminating CSPGs alone or a nerve conditioning lesion only modestly increases regeneration across the DREZ. The mechanism for this modest effect is unknown, but the default assumption has been to attribute it to myelin inhibitors which alone may be sufficient to prevent regeneration. For example, the Fawcett group largely speculated that activation of integrin produced robust functional DR regeneration due to inhibition of both myelin inhibitors and CSPGs (Tan et al., 2011; Cheah et al., 2016).

Figure 3 in our manuscript, which shows no enhanced regeneration when myelin inhibitors are genetically eliminated, is therefore the first to contradict the pharmacological results of the Frank and Mata groups. In addition, our paper is the first gene targeting study of the DREZ to simultaneously eliminate “all three major myelin inhibitors (Nogo, MAG, OMgp)”. We believe these novel convincing data will require reconsideration of the role of myelin inhibitors at the DREZ.

Figure 4 and 5, which show that the additional elimination of CSPGs only modestly enhances regeneration, represent the first combinatorial study to target both myelin inhibitors and CSPGs, and again demonstrate a robust result that will significantly impact this field, as explained above.

Figure 6 and 7, which show that simultaneously targeting both extrinsic factors (myelin inhibitors and CSPGs) and intrinsic regenerative capability (by a nerve conditioning lesion) only modestly enhances regeneration, present the results of the first such triple targeting study. These findings will stimulate new exploration of the unknown mechanisms that so powerfully prevent regeneration of even preconditioned axons in the absence of myelin inhibitors and CSPGs.

Figure 6—figure supplement 1 (newly added) shows that double conditioning lesions modestly but not robustly enhance regeneration across the DREZ in WT mice. This is the first study to report the effects of double conditioning lesions on DR regeneration across the DREZ.

Figure 1, illustrating viral labeling of proprioceptive and mechanoreceptive axons, also presents new data. Although we and other groups have recently published studies that labeled DRG axons with AAV2-GFP, this is the first presentation of evidence for its exclusive labeling of proprio/mechanoreceptive axons, but not nociceptive axons.

Figure 2 introduces a new method of wholemount analysis that we have found extremely valuable for accurate and reproducible assessment of regeneration across the DREZ. We are not aware of any other groups that have employed this method. Because the field is plagued by conflicting and inconclusive reports, this new method will help to bring a novel consistency to future analyses.

In summary, we are convinced that all the data in the manuscript are novel and significant both conceptually and technically. We again thank the reviewer for this comment and hope that this clarification will lead to fuller appreciation of the novelty and significance of our work.

2. Also, I have major concerns with most of the figures in that many do not show data for both WT and experimental (tKO, etc.). The quantified data is present, but in my opinion, images need to also be shown, especially when one of the two conditions is included. This allows the reader to assess the images themselves before accepting that the quantified data matches.

Our work compares regeneration in a total of 12 animal groups: 3 WT and 9 tKO mouse cohorts (with and without LV-CSPGs, conditioning lesion(s), and/or LV-GDNF). We placed WT at 2wpi separately in Figure 2 (which also explains the wholemount assay), because it allowed us to avoid repeatedly presenting WT images, and yet to compare reasonably large-sized images for a tKO group and immediately related tKO group(s) (i.e., Figures 3 to 9).

We do, however, understand and thank the reviewer for suggesting addition of WT images, as they will be helpful, particularly for newcomers to the DREZ. As suggested, we now include WT images in all the main figures. In addition, we have added a new Figure 2—figure supplement 1, in which wholemount and transverse images of 2 WT and 8 tKO groups are presented side-by-side. These figures will help readers to compare different groups, as the reviewer helpfully suggests.

3. Another major concern is the use of GFAP as the only marker for the boundary between the CNS and PNS. There are other makers, laminin is an example, that would give a much clearer delineation of this boundary. And in some studies/data, they don't use anything. This brings into question rigor, and I think it's essential to show this boundary given that the premise of the manuscript is to determine if axons can cross it.

We in fact used DAPI nuclear staining as an additional or alternative marker to GFAP, to identify the CNS:PNS boundary or astrocyte:PNS border. We omitted DAPI images in most figures to present other images (GFP, CGRP, CS-56, etc.) at a larger size. We regret the omission and now include DAPI images in all figures.

In most of our analyses of sections with GFP-labeled axons, we also immunolabeled CGRP- and IB4-positive axons, for simultaneous assessment of all three major subtypes of DR axons. To do this, however, we could not use GFAP or other markers that require antibody-based immunostaining: instead, we used DAPI stain as an alternative boundary label and performed 4color immunohistochemistry.

As shown in Figure 2E and the newly provided Figure 2—figure supplement 2, DAPI reveals cell nuclei that accumulate much more densely in the PNS (due to Schwann cells) than CNS, permitting us to reliably identify the CNS:PNS boundary denoted by GFAP.

To support further the rationale of using DAPI as a marker, we carried out additional experiments in which transverse sections are co-labeled with laminin, DAPI and GFAP. As shown in the new Figure 2—figure supplement 2, the boundaries denoted by DAPI, GFAP and laminin overlap closely, further validating our use of DAPI.

We also co-labeled transverse sections with these markers for the new studies of the double conditioning lesions. We observed that they all delineate the same boundary (Figure 6figure supplement 1H).

We are aware of earlier studies that used laminin to mark the boundary (e.g., Hoeber et al., 2015, 2017). We have also tried laminin and other markers such as MOG and SC2E in our earlier work. We preferred GFAP, because it has been the most commonly used, importantly including in studies that contradict the present work. Furthermore, laminin, a marker of Schwann cell basal laminae, also ubiquitously labels blood vessels, including radicular and radiculomedullary arteries at or near the boundary, as well as the entire pial covering of the spinal cord (Figure 2—figure supplement 2). This labeling can obscure precise delineation of the boundary. Likewise, peripheral invasion of astrocytic processes after root injury makes GFAP an imperfect marker. Nevertheless, peripheral invasion is not extensive, and newly invading astrocytic processes are readily distinguished from parental astrocytes (e.g., Figure 2—figure supplement 2; arrows in B’’). Accordingly, the boundaries delineated by DAPI, GFAP and laminin are closely, if not precisely, overlapping.

Lastly, we would like to emphasize that the DREZ is the zone, rather than a line, where axons stop. WT axons cross the boundary (i.e., astrocyte:PNS border marked by GFAP, DAPI or laminin) and then stop. We considered axons that reached “>100µm” from the boundary as having penetrated “the DREZ” (not the boundary), because WT axons can reach that far before they are arrested (Figure 2E and 2F, and also see Di Maio et al., 2011). Accordingly, if regeneration is indeed robustly enhanced, numerous axons should extend >200µm beyond the boundary. Therefore, they will be located unusually deep in the dorsal funiculus or in the dorsal horn grey matter, where they are readily identifiable without the need to “precisely” delineate the boundary.

4. The new data in this field is only represented in the final 2 figures, and it is preliminary. Additional studies describing the possible axon branching, etc., would make these data stronger.

As explained above (comment #1), we respectfully contend that the first 7 figures also present novel results from a series of new experiments: co-eliminating myelin inhibitors and CSPGs fails to promote robust regeneration across the DREZ of even conditioning lesioned axons. The final 2 figures extend the main findings by showing that co-eliminating these inhibitors, which could not enhance regeneration of conditioning lesioned axons, markedly and synergistically enhances regeneration of GDNF-stimulated axons across the DREZ. This result highlights the “moderate” role of these inhibitors at the DREZ, as detailed below in the response to reviewer 3’s comment #2, and the importance of sufficiently enhancing neuron intrinsic growth potential. We submit that both wholemount and section analyses clearly demonstrate these findings, as does reviewer #3: these results “show… convincingly that additional removal of inhibitors allowed for more lengthy regeneration within the CNS compartment”.

Our data indeed suggest that co-eliminating myelin inhibitors and CSPGs enhances intraspinal regeneration by stimulating sprouting, branching and extension of regenerating DR axons. First, we selectively labeled DR axons “regenerating” from injured roots. As shown in Figure 9G, the number of regenerating axons crossing the DREZ (i.e., at 100 µm) is greater than the number of regenerating axons that entered the DREZ (i.e., at the boundary at 0 µm). Furthermore, the number of GDNF-stimulated axons crossing the DREZ at 100 µm in “WT mice” is markedly increased in tKO and ChABC-expressed tKO mice. Thus, it appears that eliminating myelin inhibitors alone or additional CSPG removal elicited more lengthy regeneration by promoting considerable sprouting or branching of regenerating axons at the DREZ and likely also deeper in the CNS (i.e., “regenerative sprouting”).

Second, enhanced sprouting/branching was also demonstrated by wholemounts exhibiting densely populated GFP+ axons in the CNS compartment, past the astrocyte:PNS border (Figure 9C; arrows). Third, wholemounts showed numerous long axons projecting rostrally (Figure 9C; arrowheads), further evidence that eliminating these inhibitors promoted extension of regenerating axons. We now include these points in the revision. Also, please see below our responses to reviewer #2, who had a similar question (comment #2).

Reviewer #2:The reasons why axon regeneration into the spinal cord after a dorsal root crush are still not fully understood. In this manuscript, Zhai et al., set out to determine the role of myelin inhibitors and chondroitin sulfate proteoglycans (CSPG) in limiting axon regeneration across the dorsal root entry zone (DREZ) after a complete dorsal root crush injury. To do this, they performed L4 and L5 dorsal root crush injuries in transgenic mice that lacked the myelin proteins Nogo, MAG, and OMgp (triple knockout). In some of these animals, they injected a lentivirus for chondroitinase ABC to digest CSPGs. They also performed conditioning lesions and expression of GDNF with lentivirus. To identify axons, mainly proprioceptive and mechanoreceptive, they injected AAV2-GFP into the L4 and L5 dorsal root ganglia.Interestingly, they found that the triple knockout did not result in enhanced axon regeneration into the spinal cord. Expression of ChABC or the conditioning lesion modestly improved the axon regeneration in the triple knockout animals. There did not appear to be a synergistic effect, as the combination of the conditioning lesion and ChABC did not increase axon regeneration further. Expression of GDNF did increase growth into the spinal cord in both WT and the triple knockout animals.The manuscript is well written, and the data are clearly presented. There are a few comments to strengthen the authors' conclusions:1. This group previously showed that, after a dorsal root crush, the axons form synaptic contacts on NG2 cells that was thought to limit axon regrowth. Was the degree of this impacted in any of the groups in which they observed enhanced growth into the spinal cord? The authors allude to a synapse-based inhibitory mechanism in the discussion and refer to a manuscript that is in preparation. However, examining the number of synapses formed by these axons in the difference conditions would strengthen the data here.

We thank the reviewer for the interesting suggestion. We have previously shown that some axons stopped at the DREZ form presynaptic terminals with unknown postsynaptic cells (Di Maio et al., 2011). In a subsequent perspective article, we then “speculated” that the postsynaptic cells might be NG2 cells (Son, 2015). Despite these speculations, the identity of the postsynaptic cells is completely unknown. This will be the focus of a manuscript that we have in preparation. Moreover, NG2 cells may not be the sole mechanism inducing aberrant synapses and not all axons may stop by forming synapses (Son, 2015).

In addition, neuron-NG2 synapses have been identifiable ultrastructurally, but not immunohistochemically, making quantitative and qualitative analyses challenging. Although markers of vesicles or active zones can be used to label presynaptic terminals, we have found that they do not reliably indicate formation of the aberrant synapses at the DREZ. Accordingly, in our opinion, it would be premature and beyond the scope of the present work to investigate the degree to which aberrant synapse formation impacted the limited or enhanced regeneration by our various interventions. Instead, we have now clarified our discussion about the mechanisms that could prevent most axons from regenerating across the DREZ, even in the absence of myelin inhibitors and CSPGs and after a nerve conditioning lesion.

2. The GDNF data are interesting. The authors conclude that GDNF enabled "intraspinal penetration and regeneration". However, as presented, it is not clear that the increase in axons within the cord is due to more neurons extending an axon across the DREZ with GDNF or sprouting of axons from the same number of neurons that had already crossed the DREZ? Either would lead to more axons within the cord but by different mechanisms.

Consistent with earlier reports (Ramer et al., 2000; Kelamangalath et al., 2015, Han et al., 2017), and as shown in Figure 9G (see reviewer 1’s comment #4), intraspinally expressed GDNF increased the number of axons extending across the DREZ (i.e., measured at 100µm) ~40% in WT mice. Axons crossing the DREZ increased even more dramatically in GDNF expressed tKO (~130%) and GDNF-and ChABC-expressed tKO (~150%). The fact that the number of axons crossing the DREZ at 100µm is significantly greater than the number of axons that entered the DREZ at 0µm in tKO and ChABC-expressed tKO suggests that GDNF markedly elicited regenerative sprouting of parental DR axons in the absence of myelin inhibitors and/or CSPGs, increasing the number of axons extending across the DREZ and deeper within the spinal cord (please see the response to reviewer 1’s comment #4 for more discussion).

In our opinion, more important aspect of the study is that, although both a nerve conditioning lesion and GDNF are known to increase intrinsic growth potential, we found that elimination of myelin inhibitors and CSPGs enhances regeneration of GDNF-stimulated, but not conditioning lesioned DR axons, across the DREZ. These data therefore emphasize the importance of enhancing intrinsic growth potential sufficiently robustly. Another important implication of these data is that myelin inhibitors and CSPGs play a moderate role in restraining axon outgrowth at the DREZ. Although combined elimination of these inhibitors only modestly enhanced regeneration, even after conditioning lesion, their absence, in combination with GDNF, enabled many more axons to extend across the DREZ. Thus, these data suggest the existence of other inhibitory mechanism(s) that can potently prevent most axons from crossing the DREZ in the absence of myelin inhibitors and CSPGs. The basis of this inhibitory mechanism needs to be identified, so that we can determine if removing it alone will elicit robust penetration through the DREZ and if the same mechanism also restrains axon outgrowth in deeper portions of the CNS. As suggested, we have now clarified and amplified our discussion of the mechanisms and implications of the markedly enhanced intraspinal regeneration in GDNF- and ChABC expressed tKO mice.

3. The authors conclude that the absence of CSPG is insufficient to enable axons to cross the DREZ. Their data do not support this claim, as they did see more axons within several hundred microns in the spinal cord when ChABC was expressed (Figure 4). While it is apparent that CSPGs are not the only mechanism limiting growth across the DREZ, as discussed by the authors, their data do support that CPSGs play a role in limiting growth into the spinal cord. The conclusions should be modified accordingly.

We did not mean to imply that CSPGs (and myelin inhibitors) have no role and have clarified our statements as suggested. In addition, we changed the previous title which was somewhat misleading. Our data indeed indicate that both myelin inhibitors and CSPGs restrain axon outgrowth at the DREZ, as discussed above for the effects of GDNF (comment #2). Our conclusion is that myelin inhibitors and CSPGs are unlikely to play a “major” role, which contradicts one of the key assumptions in the field (please see the response to reviewer 1’s comment #1). We reached this conclusion because even co-elimination of myelin inhibitors and CSPGs enabled only about ~10% of axons to cross the DREZ and extend ~100µm past the DREZ (Figure 4F). Moreover, prolonged elimination (for a month after injury) did not significantly increase the number of axons extending further across the DREZ (Figure 5K). Therefore, our data suggest that other inhibitory mechanism(s) are more decisive in restraining most axons (>90%) at the DREZ in the absence of myelin inhibitors and CSPGs. Also, we believe that our finding of enhanced regeneration with GDNF but not after conditioning lesion further support our conclusion that myelin inhibitors and CSPGs play a “moderate” role, as detailed below in the response to reviewer 3’s comment #2.

Reviewer #3:The authors have described a rigorous set of experiments where they systematically removed singly or in combination, purported inhibitors of axon regeneration as well as the further addition of a single crush conditioning lesion (to promote the intrinsic capacity for growth) in order to attempt to promote regeneration of adult sensory axons across the DREZ. In the end, they found that only GDNF treatment (as has already been documented) of the DREZ (but not removal of myelin or CSPG inhibitors nor crush conditioning) was sufficient to get axons across the barrier. However, it was shown convincingly that additional removal of inhibitors allowed for more lengthy regeneration within the CNS compartment. In general, this is a well-documented series of experiments that sheds new light on the mechanisms that are at work in curtailing sensory axon regeneration at one of the most inhibitory regeneration boundaries within the nervous system, the DREZ. I have made a variety of suggestions for revisions of the text in order to soften, at least a bit, the sometimes overly zealous statements about CSPGs and myelin inhibitors. There is also a suggestion about critically altering the discussion as well as the possibility of adding a series of additional experiments (although, following the eLife guidelines, the additional experiments are deemed not to be absolutely required). My comments and suggestions for rewriting are listed in order below.Pg 3. This potent blockade is surprising…the growth potential of a relatively small number of DRG neurons to penetrate a glial scar…Pg 3. CSPG removal, however, when combined with chronic, inflammation induced conditioning lesions or neurotrophic factors, has significantly enhanced intraspinal regeneration of DR axons (Steinmetz et al., 2005; Wu et al., 68 2016; Guo et al., 2019).Pg. 3 These reports have supported a contributing inhibitory role of CSPGs by suggesting that removing CSPGs alone only modestly increases regeneration presumably because myelin inhibitors are sufficient to stop axons at the DREZ.Pg. 3 Despite the long-held view by some attributing…. to myelin inhibitors by themselves.Pg.3 …how robustly unconditioned or lesion induced conditioned DR axons regenerate….Pg. 4 …CSPGs and addition of a trauma induced conditioning lesion,….Pg 4 Thus, myelin inhibitors and CSPGs individually are not crucial factors….Pg 8 …that by themselves might be capable of arresting axons at the DREZ.Pg 8 … due to redundant and potent inhibition by other inhibitorsFurther discussion about enhanced axons at the DREZ when lenti ch'ase was delivered is warranted. Papers of David Muir are relevant here.Pg 10 Instead, these findings suggest that neither myelin inhibitors nor CSPGs that reside in the vicinity of the DREZ after injury are, by themselves, sufficiently potent to prevent regeneration of DR axons at the DREZ.Pg 10 DR axons fail to penetrate the DREZ in Nogo/MAG/OMgp tKO even after a nerve crush conditioning lesionPg 11 Therefore, removal of myelin associated inhibitors does not enable even crush conditioned axons to regenerate beyond the DREZ.

We thank the reviewer and have amended all the statements as suggested. We also mentioned enhanced axons at the DREZ and papers of David Muir.

Somewhere In the discussion I think it is critical for the authors to compare their single lesion induced conditioning results to those of Steinmetz, et al., 2005 who used week-long zymosan conditioning to induce a maximally robust conditioning lesion response. The authors should note that in Steinmetz et al., in vitro, a single surgical crush conditioning lesion (like that used in the present experiments) did not allow for regeneration of DRG axons across an inhibitory CSPG gradient (see Figure 1 in Steinmetz et al., 2005). It would have been interesting if the authors had used a double conditioning lesion paradigm (as in Neumann et al., PNAS 2005) to see if maximal surgical conditioning could enhance regeneration after elimination of one or both of the other purported axon growth inhibitors. If the authors are willing to perform the double conditioning experiments then this could enhance the paper but it is not being suggesting that this is essential. However, the lack of double conditioning experiments does somewhat dampen enthusiasm for the paper. It is being suggested that, at a minimum, the authors surely should discuss the possibility that their single conditioning paradigm is not optimal.

We completely agree with the reviewer that the single nerve crush paradigm is not optimal, and that chemical (e.g., zymogen), genetic (e.g., caBRAF, caBRAF+PTEN) and double nerve lesion induced conditioning can enhance neuron intrinsic growth capacity more strongly. We chose single nerve crush because it has been the most used method for pre-conditioning DRG neurons. More importantly, by choosing “suboptimal” conditioning, we sought to evaluate how powerfully myelin inhibitors and CSPGs impede regeneration across the DREZ. We reasoned that, if they indeed play a “major” role in arresting axons at the DREZ, then their simultaneous removal should be sufficient to drive “robust” regeneration of “even suboptimally” conditioned axons through the DREZ. On the other hand, if their role is “moderate”, as we found in the present study, then enhancing intrinsic growth potential might be essential for robust penetration. In fact, we found that the latter is the case by comparing the effects of single nerve crush with those of supplemental expression of GDNF in Nogo/MAG/OMgp tKO mice treated with LV-ChABC.

We chose GDNF over zymogen or caBRAF (O’Donovan et al., 2014), because greater enhancement of intrinsic capacity is not necessarily beneficial for evaluating the efficacy of these inhibitors in arresting axons at the DREZ: too many axons extend across the DREZ and further deeper in the spinal cord without even eliminating inhibitors. In addition, we observed that zymogen treated DRG neurons progressively die in mice, likely due to severe inflammation (Kim et al., unpublished), and that supplemental genetic conditioning of caBRAF in tKO is challenging.

Because we agree with the reviewer that single nerve crush is not optimal in mechanically conditioning DRG neurons, we have amended our statements that might have implied that it is. We are also grateful for the suggestion of the double conditioning lesion experiment. Neumann et al. observed that the double lesion enhanced regeneration of DC axons after spinal cord injury, but no previous study examined its effects on DR regeneration. We therefore decided to determine how potently a double conditioning lesion would enhance regeneration of DR axons across the DREZ in WT mice. To do this, we followed the double conditioning paradigm of Neumann et al. and repetitively “transected” sciatic nerves in WT mice. Our data shows, as presented in the new Figure 6—figure supplement 1, that the double conditioning lesion enhances regeneration of DR axons across the DREZ, to a level comparable or slightly better than a single crush lesion in Nogo/MAG/OMgp tKO (Figure 6—figure supplement 1J). However, it is far less efficacious than GDNF in enabling axons to cross the DREZ. Therefore, we will expect to observe substantially more axons extending across the DREZ when the double conditioning lesion is combined with co-eliminating myelin inhibitors and CSPGs, but substantially fewer than when GDNF treatment is combined with co-eliminating these inhibitors.

We consider that this new data, presented as a new subsection of the Result, improves the impact of the paper by demonstrating that even the double conditioning lesion paradigm fails to drive “robust” intraspinal regeneration of DR axons. These results also imply that chemical or genetic conditioning may increase intrinsic growth potential substantially higher than a mechanical nerve conditioning lesion. We have now added these points in the Discussion.

Pg 17 When the authors discuss the aberrant synapse hypothesis to explain regeneration failure at the DREZ they should also, of course, cite and discuss the published work by Busch et al., J Neurosci 2010 and Filous et al., J Neurosci 2014, who described this phenomenon in the CNS.

We now include and discuss these publications. We previously included only those references that proposed the aberrant synapse formation hypothesis for the DREZ; we intend a fuller discussion in the manuscript in preparation, which will be focused on NG2 cells.